# A Bi-metric Framework for Efficient Nearest Neighbor Search

## Abstract

We propose a new "bi-metric" framework for designing nearest neighbor data structures. Our framework assumes two dissimilarity functions: a *ground-truth* metric that is accurate but expensive to compute, and a *proxy* metric that is cheaper but less accurate. In both theory and practice, we show how to construct data structures using only the proxy metric such that the query procedure achieves the accuracy of the expensive metric, while only using a limited number of calls to both metrics. Our theoretical results instantiate this framework for two popular nearest neighbor search algorithms: DiskANN and Cover Tree. In both cases we show that, as long as the proxy metric used to construct the data structure approximates the ground-truth metric up to a bounded factor, our data structure achieves arbitrarily good approximation guarantees with respect to the ground-truth metric. On the empirical side, we apply the framework to the text retrieval problem with two dissimilarity functions evaluated by ML models with vastly different computational costs. We observe that for almost all data sets in the MTEB benchmark, our approach achieves a considerably better accuracy-efficiency tradeoff than the alternatives, such as re-ranking.

## 1 Introduction

Similarity search is a versatile and popular approach to data retrieval. It assumes that the data items of interest (text passages, images, etc.) are equipped with a distance function, which for any pair of items estimates their similarity or dissimilarity[1]. Then, given a "query" item, the goal is to return the data item that is most similar to the query. From the algorithmic perspective, this approach is formalized as the nearest neighbor search (NN) problem: given a set of $n$ points $P$ in a metric space $(X, D)$, build a data structure that, given any query point $q \in X$, returns $p \in P$ that minimizes $D(p, q)$. In many cases, the items are represented by high-dimensional feature vectors and $D$ is induced by the Euclidean distance between the vectors. In other cases, $D(p, q)$ is computed by a dedicated procedure given $p$ and $q$ (e.g., by a cross-encoder).

Over the last decade, mapping data items to feature vectors, or estimation of similarity between pairs of data items, is often done using ML models. (In the context of text retrieval, the first task is achieved by constructing bi-encoders Karpukhin et al. (2020); Neelakantan et al. (2022); Gao et al. (2021b); Wang et al. (2024), while the second task uses cross-encoders Gao et al. (2021a); Nogueira et al. (2020); Nogueira & Cho (2020)). This creates efficiency bottlenecks, as high-accuracy models are often larger and slower, while cheaper models do not achieve the state-of-the-art accuracy. Furthermore, high-accuracy models are often proprietary and accessible only through a limited interface at a monetary cost. This motivates studying "the best of both worlds" solutions which utilize many types of models to achieve favorable tradeoffs between efficiency, accuracy and flexibility.

One popular method for combining multiple models is based on "re-ranking" Liu et al. (2009). It assumes two models: one model evaluating the metric $D$, which has high accuracy but is less efficient; and another model computing a "proxy" metric $d$, which is cheap but less accurate. The algorithm uses the second model ($d$) to retrieve a large (say, $k = 1000$) number of data items with the highest similarity to the query, and then uses the first model ($D$) to select the most similar items. The hyperparameter $k$ controls the tradeoff between the accuracy and efficiency. To improve the

---

[1]To simplify the presentation, throughout this paper we assume a dissimilarity function.

efficiency further, the retrieval of the top-$k$ items is typically accomplished using approximate nearest neighbor data structures. Such data structures are constructed for the proxy metric $d$, so they remain stable even if the high-accuracy metric $D$ undergoes frequent updates.

Despite its popularity, the re-ranking approach suffers from several issues:

1. The overall accuracy is limited by the accuracy of the cheaper model. To illustrate this phenomenon, suppose that $D$ defines the "true" distance, while $d$ only provides a "$C$-approximate" distance, i.e., that the values of $d$ and $D$ for the same pairs of items differ by at most a factor of $C > 1$. Then the re-ranking approach can only guarantee that the top reported item is a $C$-approximation, namely that its distance to the query is at most $C$ times the distance from the query to its true nearest neighbor according to $D$. This occurs because the first stage of the process, using the proxy $d$, might not retain the most relevant items.

2. Since the set of the top-$k$ items with respect to the more accurate model depends on the query, one needs to perform at least a linear scan over all $k$ data items retrieved using the proxy metric $d$. This computational cost can be reduced by decreasing $k$, but at the price of reducing the accuracy.

**Our results**   We show that, in both theory and practice, it is possible to combine cheap and expensive models to achieve approximate nearest neighbor data structures that inherit the accuracy of expensive models while significantly reducing the overall computational cost. Specifically, we propose a *bi-metric framework* for designing nearest neighbor data structures with the following properties:

- The algorithm for creating the data structure uses only the proxy metric $d$, making it efficient to construct,

- The algorithm for answering the nearest neighbor query leverages both models, but performs only a sub-linear number of evaluations of $d$ and $D$,

- The data structure achieves the accuracy of the expensive model.

For a more formal description of the framework, see Preliminaries (Section 2).

The simplest approach to constructing algorithms that conform to our framework is to *construct the data structure using the proxy metric $d$, but answer queries using the accurate metric $D$*; we also propose more complex solutions that have better practical performance. We note our approach is quite general, and is applicable to any approximate nearest neighbor data structure that works for general metrics. Our *theoretical* study analyzes the simple approach when applied to two popular algorithms: DiskANN Jayaram Subramanya et al. (2019) and Cover Tree Beygelzimer et al. (2006), under natural assumptions about the intrinsic dimensionality of the data, as in Indyk & Xu (2023). Perhaps surprisingly, we show that despite the fact that only the proxy $d$ is used in the indexing stage, the query answering procedure essentially retains the accuracy of the ground truth metric $D$.

Formally, we show the following theorem statement. We use $\lambda_d$ to refer to the doubling dimension with respect to metric $d$ (a measure of intrinsic dimensionality, see Definition 2.2).

**Theorem 1.1** (Summary, see Theorems 3.4 and C.3). *Given a dataset $X$ of $n$ points, $\mathtt{Alg} \in \{DiskANN, Cover\ Tree\}$, and a fixed metric $d$, let $S_{\mathtt{Alg}}(n, \varepsilon, \lambda_d)$ and $Q_{\mathtt{Alg}}(\varepsilon, \lambda_d)$ denote the space and query complexity respectively of the standard datastructure for $\mathtt{Alg}$ which reports a $1 + \varepsilon$ nearest neighbor in $X$ for any query (all for a fixed metric $d$).*

*Consider two metrics $d$ and $D$ satisfying Equation 1. Then for any $\mathtt{Alg} \in \{DiskANN, Cover\ Tree\}$, we can build a corresponding datastructure $\mathcal{D}_{\mathtt{Alg}}$ on $X$ with the following properties:*

    *1. When constructing $\mathcal{D}_{\mathtt{Alg}}$, we only access metric $d$,*

    *2. The space used by $\mathcal{D}_{\mathtt{Alg}}$ can be bounded by $\tilde{O}(S_{\mathtt{Alg}}(n, \varepsilon/C, \lambda_d))^2$,*

    *3. Given any query $q$, $\mathcal{D}_{\mathtt{Alg}}$ invokes $D$ at most $\tilde{O}(Q_{\mathtt{Alg}}(\varepsilon/C, \lambda_d))$ times,*

    *4. $\mathcal{D}_{\mathtt{Alg}}$ returns a $1 + \varepsilon$ approximate nearest neighbor of $q$ in $X$ under metric $D$.*

---

[2]$\tilde{O}$ hides logarithm dependencies in the aspect ratio.

The proof of the theorem crucially uses the properties of the underlying data structures. It is an interesting open direction to determine if our bi-metric framework can be theoretically instantiated for other popular nearest neighbor algorithms, such as those based on locality sensitive hashing.

To demonstrate the *practical* applicability of the bi-metric framework, we apply it to the text retrieval problem. Here, the data items are text passages, and the goal is to retrieve a passage from a large collection that is most relevant to a query passage. We instantiated our framework with the DiskANN algorithm, using a high-quality "SFR-Embedding-Mistral" model Meng et al. (2024) to define $D$, and a lower-quality "bge-micro-v2" model AI (2023) to define $d$. Both metrics $d(p, q)$ and $D(p, q)$ are induced by the Euclidean distance between the embeddings of $p$ and $q$ using the respective models. The sizes of the two models differ by 3 orders of magnitude, making $D$ much more expensive to evaluate than $d$. Specifically, in our experiments, embedding a single passage takes 0.00043 seconds when using bge-micro-v2 and 0.13 seconds when using SFR-Embedding-Mistral, making the second model >300 times slower.

We evaluated the retrieval quality of our approach on a benchmark collection of 15 MTEB retrieval data sets Thakur et al. (2021), comparing it to the re-ranking approach, which retrieves the closest data items to the query with respect to $d$ and re-ranks using $D$. We observe that for almost all data sets, our approach achieves a considerably better accuracy-efficiency tradeoff than re-ranking. In particular, for several data sets, such at HotpotQA Yang et al. (2018), our approach achieves state-of-the-art retrieval accuracy using up to **4x** fewer evaluations of the expensive model.

## 1.1 RELATED WORK

As described in the introduction, a popular method for utilizing a cheap metric $d$ and expensive metric $D$ in similarity search is based on "filtering" or "re-ranking". The idea is to use $d$ to construct a (long) list of candidate answers, which is then filtered using $D$. It is a popular approach in many applications, including recommendation systems Liu et al. (2022) and computer vision Zhong et al. (2017). Due to the popularity of this method, we use it as a baseline in our experiments.

In addition to the re-ranking method, multiple other papers proposed different methods for combining accurate and cheap metrics to improve similarity search and related problems. We discuss those papers in more detail below. We note that, with the exception of Moseley et al. (2021); Silwal et al. (2023); Bateni et al. (2024), those methods do not appear to come with provable correctness or efficiency guarantees, or generally applicable frameworks (in contrast to the proposal in this paper). Furthermore, the three aforementioned papers Moseley et al. (2021); Silwal et al. (2023); Bateni et al. (2024) focus on various forms of clustering, not on similarity search. The paper Moseley et al. (2021) is closest to our work, as it uses approximate nearest neighbor as a subroutine when computing the clustering. However, their algorithm only achieves the (lower) accuracy of the cheaper model, while our algorithms retains the (higher) accuracy of the expensive one.

There are also several other empirical works on similarity search that combine cheap and expensive metrics, none of which fully capture our framework to the best of our knowledge. The aforementioned paper Jayaram Subramanya et al. (2023) describes (in section 3.1) an optimization which uses the ground truth metric $D$ during the indexing phase, and proxy metric $d$ during the search phase. In contrast, our framework uses $D$ during the search phase and $d$ during indexing. This difference seems crucial to our ability of providing strong approximation guarantees for the reported points, which are not limited by the distortion $C$ between $d$ and $D$. In another paper Chen et al. (2023), the authors use the proxy metric $d$ obtained by "sketching" $D$ during the query answering phase, in order to prune some points from the search queue without resorting to computing $D$. However, the data structure index is still constructed using the expensive metric $D$, as opposed the proxy metric $d$ as in our framework, which makes preprocessing more expensive in terms of space and time. Finally, Morozov & Babenko (2019) present a method for constructing a similarity graph with respect to an approximate distance function derived from a complex one; during the query phase the graph is explored using a more complex relevance function. However, their algorithm uses specific proxy metric derived from the expensive one; in contrast, our framework allows arbitrary distance functions $d$ and $D$, as long as the distortion $C$ between them is bounded.

**Graph-based algorithms for similarity search** The algorithms studied in this paper rely on graph-based data structures for (approximate) nearest neighbor search. Such data structures work

for general metrics, which, during the pre-processing, are approximated by carefully constructed graphs. Given the graph and the query point, the query answering procedure greedily searches the graph to identify the nearest neighbors. Graph-based algorithms have been extensively studied both in theory Krauthgamer & Lee (2004); Beygelzimer et al. (2006) and in practice Fu et al. (2019b); Jayaram Subramanya et al. (2019); Malkov & Yashunin (2018); Harwood & Drummond (2016). See Clarkson et al. (2006); Wang et al. (2021) for an overview of these lines of research.

## 2 PRELIMINARIES

**Nearest neighbor search** We first consider the standard formulation of *exact* nearest neighbor search. Here, we are given a set of points $P$, which is a subset of the set of all points $X$ (e.g., $X = \mathbb{R}^d$). In addition, we are given access to a metric function $D$ that, for any pair of points $p, q \in X$ returns the dissimilarity between $p$ and $q$. The goal of the problem is to build an index structure that, given a *query* point $q \in X$, returns $p^* \in P$ such that

$$p^* = \arg\min_{p \in P} D(q, p).$$

The formulation is naturally extended to more general settings, such as:

- $(1 + \varepsilon)$-approximate nearest neighbor search, where the goal is to find any $p^* \in P$ such that
$$D(q, p^*) \leq (1 + \varepsilon) \min_{p \in P} D(q, p).$$

- $k$-nearest neighbor search, where the goal is to find the set of $k$ nearest neighbors of $q$ in $P$ with respect to $D$. If the algorithm returns a set $S'$ of $k$ points that is different than the set $S$ of true $k$ nearest neighbor, the quality of the answer is measured by computing the Recall rate or NDCG score Järvelin & Kekäläinen (2002).

**Bi-metric framework:** In our framework, we assume that we are given *two* metrics over $X$:

- The *ground truth* metric $D$, which for any pair of points $p, q \in X$ returns the "true" dissimilarity between $p$ and $q$. The metric $D$ plays the same role as in the standard nearest neighbor search problem.

- The *proxy* metric $d$, which provides a cheap approximation to the ground truth metric.

Throughout the paper, we think of $D$ as being 'expensive' to evaluate, while $d$ as the cheaper, but noisy, proxy.

We assume that the algorithm for constructing the data structure has access to the proxy metric $d$, but *not* to the ground truth metric $D$. The algorithm for answering a query $q$ has access to *both* metrics. However, the complexity of the query-answering procedure is measured by counting only the number of evaluations of the expensive metric $D$.

As described in the introduction, the above formulation is motivated by the following considerations:

- Computing all embeddings using the expensive model $D$ (e.g. SFR-Embedding-Mistral (Meng et al., 2024)) requires lots of time and space. For example, it takes an A100 gpu around 196 hours to compute all embeddings of 5 million passages from the HotpotQA dataset and these embeddings occupy 83GB of disk storage. Meanwhile, using the cheap model $d$ (e.g. bge-micro (AI, 2023)), computing these embeddings only takes 0.62 hours and 7GB of disk storage. As a comparison, the graph index size of 5 million points occupies roughly 1GB of disk storage.

- Evaluating ground truth metric $D$ during query answering time is also very expensive, due to factors such as model size or monetary costs associated with querying proprietary models. Therefore, our cost model for the query answering procedure only accounts for the number of such evaluations.

- In applications that use similarity search data structures in model training, the metric $D$ can change after each model update, necessitating re-computing embeddings and the search index over the entire database. Since this is expensive, some works (e.g., (Borgeaud et al., 2022)) freeze the parts of the model that compute embeddings to avoid the computational cost of updating the data structure. Our framework offers an alternative approach, where one constructs a stable index for a proxy $d$ using frozen embeddings, but uses the up-to-date model to compute the ground-truth metric $D$ when answering nearest neighbor queries.

**Assumptions about metrics:** Clearly, if the metrics $d$ and $D$ are not related to each other, the data structure constructed using $d$ alone does not help with the query retrieval. Therefore, we assume that the two metrics are related through the following definition.

**Definition 2.1.** Given a set of $n$ points $P$ in a metric space $X$ and its distance function $D$, we say the distance function $d$ is a $C$-approximation[3] of $D$ if for all $x, y \in X$,

$$d(x,y) \leq D(x,y) \leq C \cdot d(x,y). \tag{1}$$

For a fixed metric $d$ and any point $p \in X$, radius $r > 0$, we use $B(p, r)$ to denote the ball with radius $r$ centered at $p$, i.e. $B(p, r) = \{q \in X : d(p, q) \leq r\}$. In our paper, the notion of *doubling-dimension* is central. It is a measure of intrinsic dimensionality of datasets which is popular in analyzing high dimensional datasets, especially in the context of nearest neighbor search algorithms Gupta et al. (2003); Krauthgamer & Lee (2004); Beygelzimer et al. (2006); Indyk & Naor (2007); Har-Peled & Kumar (2013); Narayanan et al. (2021); Indyk & Xu (2023).

**Definition 2.2** (Doubling Dimension). $X$ has doubling dimension $\lambda_d$ with respect to metric $d$ if for any $p \in X$, and radius $r > 0$, $X \cap B(p, 2r)$ can be covered by at most $2^{\lambda_d}$ balls with radius $r$.

Finally, for a metric $d$, $\Delta_d$ is the aspect ratio of the input set $X$, i.e., the ratio between the diameter and the distance of the closest pair.

## 3 THEORETICAL ANALYSIS

We instantiate our *bi-metric* framework for two popular nearest neighbor search algorithms: DiskANN and Cover Tree. The goal of our bi-metric framework is to first create a data structure using the proxy (cheap) metric $d$, but solve nearest neighbor to $1 + \varepsilon$ accuracy for the expensive metric $D$. Furthermore, the query step should invoke the metric $D$ judiciously, as the number of calls to $D$ is the measure of efficiency. Our theoretical query answering algorithms do not use calls to $d$ at all.

We note that, if we treat the proxy data structure as a *black box*, we can only guarantee that it returns a $C$-approximate nearest neighbor with respect to $D$. Our theoretical analysis overcomes this, and shows that calling $D$ a sublinear number of times in the query phase (for DiskANN and Cover Tree) allows us to obtain *arbitrarily accurate* neighbors for $D$.

At a high level, the unifying theme of the algorithms that we analyze (DiskANN and Cover Tree) is that they both crucially use the concept of a *net*: given a parameter $r$, a $r$-net is a small subset of the dataset guaranteeing that every data point is within distance $r$ to the subset in the net. Both algorithms (implicitly or explicitly), construct nets of various scales $r$ which help route queries to their nearest neighbors in the dataset. The key insight is that a net of scale $r$ for metric $d$ is also a net under metric $D$, but with the larger scale $Cr$. Thus, if we construct smaller nets for metric $d$, they can also function as nets for the expensive metric $D$ (which we don't access during our data structure construction). Care must be taken to formalize this intuition and we present the details below.

We remark that the intuition we gave clearly does not generalize for nearest neighbor algorithms which are fundamentally different, such as locality sensitive hashing. For such algorithms, it is not clear if any semblance of a bi-metric framework is algorithmically possible, and this is an interesting open direction.

In the main body, we present the (simpler) analysis of DiskANN and defer the analysis of Cover Tree to Appendix C.

### 3.1 DISKANN

**Preliminaries for DiskANN.** First, some helpful background is given. In this section, we only deal with a single metric $d$. We first need the notion of an $\alpha$-shortcut reachability graph. Intuitively, it is an unweighted graph $G$ where the vertices correspond to points of a dataset $X$ such that nearby points (geometrically) are close in graph distance.

**Definition 3.1** ($\alpha$-shortcut reachability Indyk & Xu (2023)). Let $\alpha \geq 1$. We say a graph $G = (X, E)$ is $\alpha$-shortcut reachable from a vertex $p$ under a given metric $d$ if for any other vertex $q$, either

---

[3]Please see Section 4 and Figure 11 for empirical estimates of $C = D/d$.

$(p, q) \in E$, or there exists $p'$ s.t. $(p, p') \in E$ and $d(p', q) \cdot \alpha \leq d(p, q)$. We say a graph $G$ is $\alpha$-shortcut reachable under metric $d$ if $G$ is $\alpha$-shortcut reachable from any vertex $v \in X$.

The main analysis of Indyk & Xu (2023) shows that (the 'slow preprocessing version' of ) DiskANN outputs an $\alpha$-shortcut reachability graph.

**Theorem 3.2** (Indyk & Xu (2023)). *Given a dataset $X$, $\alpha \geq 1$, and fixed metric $d$ the slow preprocessing DiskANN algorithm (Algorithm 4 in Indyk & Xu (2023)) outputs a $\alpha$-shortcut reachibility graph $G$ on $X$ as defined in Definition 3.1 (under metric $d$). The space complexity of $G$ is $n \cdot \alpha^{O(\lambda_d)} \log(\Delta_d)$.*

Given an $\alpha$-reachability graph on a dataset $X$ and a query point $q$, Indyk & Xu (2023) additionally show that the greedy search procedure of Algorithm 1 finds an accurate nearest neighbor of $q$ in $X$.

**Theorem 3.3** (Theorem 3.4 in Indyk & Xu (2023)). *For $\varepsilon \in (0, 1)$, there exists an $\Omega(1/\varepsilon)$-shortcut reachable graph index for a metric $d$ with max degree $Deg \leq (1/\varepsilon)^{O(\lambda_d)} \log(\Delta_d)$ (via Theorem 3.2). For any query $q$, Algorithm 1 on this graph index finds a $(1 + \varepsilon)$ nearest neighbor of $q$ in $X$ (under metric $d$) in $S \leq O(\log(\Delta_d))$ steps and makes at most $S \cdot Deg \leq (1/\varepsilon)^{O(\lambda_d)} \log(\Delta_d)^2$ calls to $d$.*

We are now ready to state the main theorem of Section 3.1.

**Theorem 3.4.** *Let $Q_{\mathtt{DiskAnn}}(\varepsilon, \Delta_d, \lambda_d) = (1/\varepsilon)^{O(\lambda_d)} \log(\Delta_d)^2$ denote the query complexity of the standard DiskANN data structure[4], where we build and search using the same metric $d$. Consider two metrics $d$ and $D$ satisfying Equation 1. Suppose we build an $C/\varepsilon$-shortcut reachability graph $G$ using Theorem 3.2 for metric $d$, but search using metric $D$ in Algorithm 1 for a query $q$. Then:*

1. *The space used by $G$ is at most $n \cdot (C/\varepsilon)^{O(\lambda_d)} \log(\Delta_d)$.*

2. *Running Algorithm 1 using $D$ finds a $1 + \varepsilon$ nearest neighbor of $q$ in the dataset $X$ (under $D$).*

3. *On any query $q$, Algorithm 1 invokes $D$ at most $Q_{\mathtt{DiskAnn}}(\varepsilon/C, C\Delta_d, \lambda_d)$.*

To prove the theorem, we first show that a shortcut reachability graph of $d$ is also a shortcut reachability graph of $D$, albeit with slightly different parameters. See Section B for the lemma's proof.

**Lemma 3.5.** *Suppose metrics $d$ and $D$ satisfy relation (1). Suppose $G = (X, E)$ is $\alpha$-shortcut reachable under $d$ for $\alpha > C$. Then $G = (X, E)$ is an $\alpha/C$-shortcut reachable under $D$.*

*Proof of Theorem 3.4.* By Lemma 3.5, the graph $G = (X, E)$ constructed for metric $d$ is also a $O(1/\varepsilon)$ reachable for the other metric $D$. Then we simply invoke Theorem 3.3 for a $(1/\varepsilon)$-reachable graph index for metric $D$ with degree limit $Deg \leq (C/\varepsilon)^{O(\lambda_d)} \log(\Delta_d)$ and the number of greedy search steps $S \leq O(\log(C\Delta_d))$. Thus the total number of $D$ distance call bounded by $(C/\varepsilon)^{O(\lambda_d)} \log(C\Delta_d)^2 \leq Q_{\mathtt{DiskAnn}}(\varepsilon/C, C\Delta_d, \lambda_d)$. This proves the accuracy bound as well as the number of calls we make to metric $D$ during the greedy search procedure of Algorithm 1. The space bound follows from Theorem 3.2, since $G$ is a $C/\varepsilon$-reachability graph for metric $d$. □

## 4 EXPERIMENTS

We present an experimental evaluation of our approach. The starting point of our implementation is the DiskANN based algorithm from Theorem 3.4, which we engineer to optimize performance[5]. We compare it to two other methods on all 15 MTEB retrieval tasks Thakur et al. (2021).

### 4.1 EXPERIMENT SETUP

**Methods** We evaluate the following methods. $\mathcal{Q}$ denotes the query budget, i.e., the maximum number of calls an algorithm can make to $D$ during a query. We vary $\mathcal{Q}$ in our experiments.

- Bi-metric (our method): We build a graph index with the cheap distance function $d$ (we discuss our choice of graph indices in the experiments shortly). Given a query $q$, we first search for $q$'s

---

[4]I.e., the upper bound on the number of calls made to $d$ on any query

[5]Our experiments are run on 56 AMD EPYC-Rome processors with 400GB of memory and 4 NVIDIA RTX 6000 GPUs. Our experiment in Figure 1 takes roughly 3 days.

top-$\mathcal{Q}/2$ nearest neighbor under metric $d$. Then, we start a second-stage search from the $\mathcal{Q}/2$ returned vertices using distance function $D$ on the same graph index until we reach the quota $\mathcal{Q}$. We report the 10 closest neighbors seen so far by distance function $D$.

- Bi-metric (baseline): This is the standard retrieve + rerank method that is widely popular. We build a graph index with the cheap distance function $d$. Given a query $q$, we first search for $q$'s top-$\mathcal{Q}$ nearest neighbor under metric $d$. As explained below, we can assume that empirically the first step returns the *true* top-$\mathcal{Q}$ nearest neighbors under $d$. Then, we calculate distance using $D$ for all the $\mathcal{Q}$ returned vertices and report the top-10.

- Single metric: This is the standard nearest neighbor search with a single distance function $D$. We build the graph index directly with the expensive distance function $D$. Given a query $q$, we do a standard greedy search to get the top-10 closest vertices to $q$ with respect to distance $D$ until we reach quota $\mathcal{Q}$. We help this method and ignore the large number of $D$ distance calls in the indexing phase and only count towards the quota in the search phase. Note that this method doesn't satisfy our "bi-metric" formulation as it uses an extensive number of $D$ distance calls ($\Omega(n)$ calls) in index construction. However, we implement it for comparison since it represents a natural baseline, if one does not care about the prohibitively large number of calls made to $D$ during index building.

For both Bi-metric methods (ours and baseline), in the first-stage search under distance $d$, we initialize the parameters of the graph index so that empirically, it returns the true nearest neighbors under distance $d$. This is done by setting the 'query length' parameter $L$ to be 30000 for dataset with corpus size $> 10^6$ (Climate-FEVER Diggelmann et al. (2020), FEVER Thorne et al. (2018), HotpotQA Yang et al. (2018), MSMARCO Bajaj et al. (2018), NQ Kwiatkowski et al. (2019), DBPedia Hasibi et al. (2017)) and 5000 for the other datasets. Our choice of $L$ is large enough to ensure that the returned vertices are almost true nearest neighbors under distance $d$. For example, the standard parameters used are a factor of 10 smaller. We also empirically verified that the nearest neighbors returned for $d$ with such large values of $L$ corroborated with published MTEB benchmark values [6].

**Datasets**  We experiment with all of the following 15 MTEB retrieval datasets: Arguana Wachsmuth et al. (2018), ClimateFEVERDiggelmann et al. (2020), CQADupstackRetrievalHoogeveen et al. (2015), DBPediaHasibi et al. (2017), FEVERThorne et al. (2018), FiQA2018Maia et al. (2018), HotpotQAYang et al. (2018), MSMARCOBajaj et al. (2018), NFCorpusBoteva et al. (2016), NQKwiatkowski et al. (2019), QuoraRetrievalThakur et al. (2021) SCIDOCSCohan et al. (2020), Sci-FactWadden et al. (2020), Touche2020Bondarenko et al. (2020), TRECCOVIDVoorhees et al. (2021). As a standard practice, we report the results on these dataests' test split, except for MSMARCO where we report the results on its dev split.

**Embedding Models**  We select a highly ranked model "SFR-Embedding-Mistral" as our expensive model to provide groundtruth metric $D$. Meanwhile, we select three models on the pareto curve of the MTEB retrieval size-average score plot to test how our method performs under different model scale combinations. These three small models are "bge-micro-v2", "gte-small", "bge-base-en-v1.5". Please refer to Table 1 for details.

As described earlier, both metrics $d(p, q)$ and $D(p, q)$ are induced by the Euclidean distance between the embeddings of $p$ and $q$ using the respective models. The embeddings defining the proxy metric $d$ are pre-computed and stored during the pre-processing, and then used to construct the data structure. The embeddings defining the accurate metric $D$ are computed on the fly during the query processing stage. Specifically, to answer a query $q$, the algorithm first computes the embedding $f(q)$ of $q$. Then, whenever the value of $D(q, p)$ is needed, the algorithm computes $f(p)$ and evaluates $D(p, q) = \|f(q) - f(p)\|$. Thus, the cost of evaluating $D(p, q)$ is equal to the cost of embedding $p$. (In other scenarios where $D(p, q)$ is evaluated using a proprietary system over the Internet, the cost is determined by the vendor's prices and/or the network speed.).

**Nearest Neighbor Search Algorithms**  The nearest neighbor search algorithms we employ in our experiments are DiskANN (Jayaram Subramanya et al., 2019) and NSG (Fu et al., 2019a). We use standard parameter choices for both; see Section D.

---

[6]from `https://huggingface.co/spaces/mteb/leaderboard`

| Model Name | Embedding Dimension | Model Size | MTEB Retrieval Score |
|---|---|---|---|
| SFR-Embedding-Mistral Meng et al. (2024) | 4096 | 7111M | 59 |
| bge-base-en-v1.5 Xiao et al. (2023) | 768 | 109M | 53.25 |
| gte-small Li et al. (2023) | 384 | 33M | 49.46 |
| bge-micro-v2 AI (2023) | 384 | 17M | 42.56 |

Table 1: Different models used in our experiments

**Metric** Given a fixed expensive distance function quota $\mathcal{Q}$, we report the accuracy of retrieved results for different methods. We always insure that all algorithms never use more than $\mathcal{Q}$ expensive distance computations. Following the MTEB retrieval benchmark, we report the NDCG@10 score. Following the standard nearest neighbor search algorithm benchmark metric, we also report the Recall@10 score compared to the true nearest neighbor according to the expensive metric $D$.

## 4.2 EXPERIMENT RESULTS AND ANALYSIS

Please refer to Figure 1 for our results with $d$ distance function set to "bge-micro-v2" and $D$ set to "SFR-Embedding-Mistral", with the underlying graph index being DiskANN. To better focus on the convergence speed of different methods, we cut off the y-axis at a relatively high accuracy, so some curves may not start from x equals 0 if their accuracy doesn't reach the threshold. We observe that our method converges to the optimal accuracy much faster than bi-metric (baseline) and single metric in most cases. For example for HotpotQA, the NDCG@10 score achieved by the baselines for 8000 calls to $D$ is comparable to our method, using less than 2000 calls to $D$, leading to **>4x** fewer evaluations of the expensive model. This leads to substantial time savings. For example, consider our largest data set HotpotQA. The first stage of the query answering procedure (using $d$) takes only 0.37s per query $q$, while each evaluation of $D(p, q)$ during the second stage takes 0.13s; this translates into roughly 260s per query when 2000 evaluations of $D$ are used. In contrast, the baseline method requires 8000 calls to $D$, which translates into a cost of roughly 1040s per query.

This means that utilizing the graph index built for the distance function proxy to perform a greedy search using $D$ is more efficient than naively iterating the returned vertex list to re-rank using $D$ (baseline). It is also noteworthy to see that our method converges faster than "Single metric" in all the datasets except FiQA2018 and TRECCOVID, especially in the earlier stages. This phenomenon happens even if "Single metric" is allowed infinite expensive distance function calls in its indexing phase to build the ground truth graph index. This suggests that the quality of the underlying graph index is not as important, and the early routing steps in the searching algorithm can be guided with a cheap distance proxy functions to save expensive distance function calls.

Similar conclusion holds for the recall plot (see Figure 4) as well, where our method has an even larger advantage over Bi-metric (baseline) and is also better than the Single metric in most cases, except for FEVER, FiQA2018, and HotpotQA. We report the results of using different model pairs, using the NSG algorithm as our graph index, and measuring Recall@10 in Appendix D. Please see their ablation studies in Section 4.3.

Lastly, we measure the empirical value of $C$ (the relationship between $d\&D$ from (1)). For simplicity, we assumed that $d \leq D \leq C \cdot d$ for $C \geq 1$ in (1) in our theoretical bounds. This is without loss of generality by scaling, and we could have alternatively written our theorem statements by substituting $(\max_{x,y} D(x,y)/d(x,y))/(\min_{x,y} D(x,y)/d(x,y))$ for $C$. In practice, we observe that the ratio of distances $C := D/d$ is always clustered around one. For example, if we use "SFR-Embedding-Mistral" to provide the distance $D$, and "bge-micro-v2" to provide the distance $d$, then for HotpotQA, we empirically found that $99.9\%$ of $10^5$ randomly sampled pairs satisfy $0.6 \leq C \leq 1.5$. We observed the same qualitative behaviour for our other datasets; see Figure 11 in the appendix for more details.

## 4.3 ABLATION STUDIES

We investigate the impact of different components of our method. All ablation studies are run on HotpotQA dataset as it is one of the largest and most difficult retrieval dataset where the performance gaps between different methods are substantial.

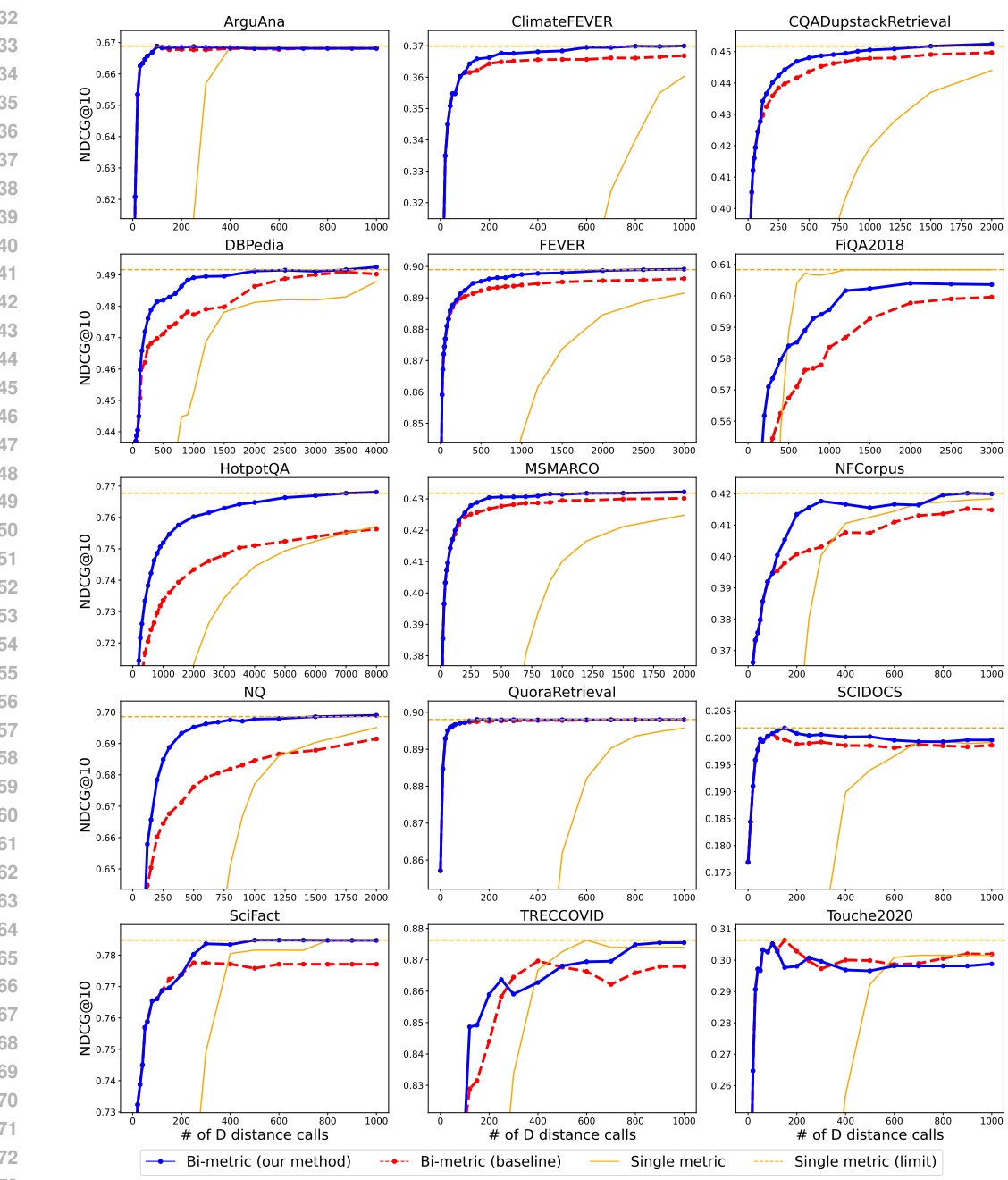

Figure 1: Results for 15 MTEB Retrieval datasets. The x-axis is the number of expensive distance function calls. The y-axis is the NDCG@10 score. The cheap model is "bge-micro-v2", the expensive model is "SFR-Embedding-Mistral", and the nearest neighbor search algorithm used is DiskANN.

**Different model pairs** Fixing the expensive model as "SFR-Embedding-mistral" (Meng et al., 2024), we experiment with 2 other cheap models from the MTEB retrieval benchmark: "gte-small" Li et al. (2023) and "bge-base" Xiao et al. (2023). These models have different sizes/capabilities, summarized in Table 1. For complete results on all 15 MTEB Retrieval datasets for different cheap models, we refer to Figures 5, 6, 7, and 8 in Appendix D. Here, we only focus on HotpotQA.

From Figure 2, we can observe that the improvement of our method is most substantial when there is a large gap between the qualities of the cheap and expensive models. This is not surprising: If the cheap model has already provided enough accurate distances, simple re-ranking can easily get to the optimal retrieval results with only a few expensive distance calls. Note that even in the latter case,

our second-stage search method still performs at least as good as re-ranking. Therefore, we believe that the ideal scenario for our method is a small and efficient model deployed locally, paired with a remote large model accessed online through API calls to maximize the advantages of our method.

**Varying neighbor search algorithms** We implement our method with another popular empirical nearest neighbor search algorithm called NSG (Fu et al., 2019b). We obtain the same qualitative behavior as DiskANN, with details given in Section D.

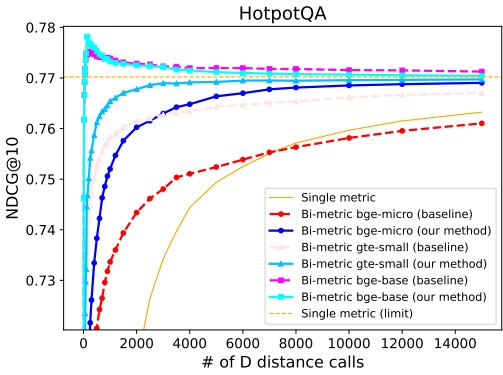 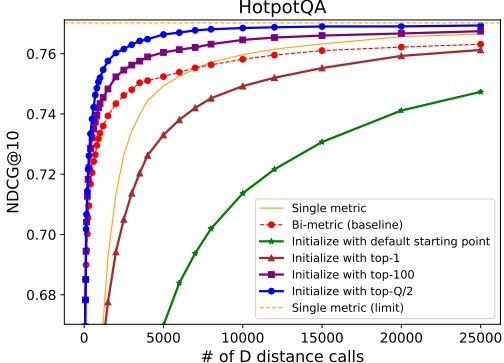

Figure 2: HotpotQA test results for different models as the distance proxy. Blue / skyblue / cyan curves represent Bi-metric (our method) with bge-micro / gte-small / bge-base models. Red / rose / magenta curves represent Bi-metric (baseline) with bge-micro / gte-small / bge-base models

Figure 3: HotpotQA test results for different search initializations for the second-stage search of Bi-metric (our method). Blue / purple / brown / green curves represent initializing our second-stage search with top-$\mathcal{Q}/2$, top-100, top-1, or the default vertex.

**Impact of the first stage search** In the second-stage search of our method, we start from multiple points returned by the first-stage search via the cheap distance metric. We investigate how varying the starting points for the second-stage search impact the final results. We try four different setups:

- Default: We start a standard nearest neighbor search using metric $D$ from the default entry point of the graph index, which means that we don't use the first stage search.
- Top-$K$ points retrieved by the first stage search: Suppose our expensive distance calls quota is $\mathcal{Q}$. We start our second search from the top $K$ points retrieved by the first stage search. We experiment with the following different choices of $K$: $K_1 = 1$, $K_{100} = 100$, $K_{\mathcal{Q}/2} = \max(100, \mathcal{Q}/2)$ (note $K_{\mathcal{Q}/2}$ is the choice we use in Figure 1).

From Figure 3, we observe that utilizing results from the first-stage search helps the second-stage search to find the nearest neighbor quicker. For comparison, we experiment with initializing the second-stage search from the default starting point (green), which means that we don't need the first-stage search and only use the graph index built from $d$ (cheap distance function). The DiskANN algorithm still manages to improve as the allowed number of $D$ distance calls increases, but it converges the slowest compared to all the other methods.

Using multiple starting points further speeds up the second stage search. If we only start with the top-1 point from the first stage search (brown), its NDCG curve is still worse than Bi-metric (baseline, red) and Single metric (orange). As we switch to top-100 (purple) or top-$\mathcal{Q}/2$ (blue) starting points, the NDCG curves increase evidently.

We provide two intuitive explanations for these phenomena. First, the approximation error of the cheap distance function doesn't matter that much in the earlier stage of the search, so the first stage search with the cheap distance function can quickly get to the true 'local' neighborhood without any expensive distance calls, thus saving resource for the second stage search. Second, the ranking provided by the cheap distance function is not accurate because of its approximation error, so starting from multiple points should give better results than solely starting from the top few, which also justifies the advantage of our second-stage search over re-ranking.

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

# A  QUERY ALGORITHM OF DISKANN

---

**Algorithm 1** DiskANN-GreedySearch($q, d$)

---

1: **Input**: Graph index $G = (X, E)$, distance function $d$, starting point $s$, query point $q$
2: **Output**: visited vertex list $U$
3: $s \leftarrow$ an arbitrary starting point in $X$
4: $A \leftarrow \{s\}$
5: $U \leftarrow \varnothing$
6: **while** $A \setminus U \neq \varnothing$ **do**
7:     $v \leftarrow \text{argmin}_{v \in A \setminus U} \, d(x_v, q)$
8:     $A \leftarrow A \cup Neighbors(v)$                                            $\triangleright$ Neighbors in $G$
9:     $U \leftarrow U \cup v$
10:     **if** $|A| > 1$ **then**
11:         $A \leftarrow$ closest vertex to $q$ in $A$
12: sort $U$ in increasing distance from $q$
13: **return** $U$

---

# B  OMITTED DISKANN PROOFS

*Proof of Lemma 3.5.* Let $(p, q)$ be a pair of distinct vertices such that $(p, q) \notin E$. Then we know that there exists a $(p, p') \in E$ such that $d(p', q) \cdot \alpha \leq d(p, q)$. From relation (1), we have $\frac{1}{C} \cdot D(p', q) \cdot \alpha \leq d(p', q) \cdot \alpha \leq d(p, q) \leq D(p, q)$, as desired. □

# C  ANALYSIS OF COVER TREE

We now analyze Cover Tree under the bi-metric framework. First, some helpful background is presented below.

### C.0.1  PRELIMINARIES FOR COVER TREE

The notion of a cover is central.

**Definition C.1** (Cover). A $r$-cover $\mathcal{C}$ of a set $X$ given a metric $d$ is defined as follows. Initially $\mathcal{C} = \emptyset$. Run the following two steps until $X$ is empty.

    1. Pick an arbitrary point $x \in X$ and remove $B(x, r) \cap X$ from $X$.

    2. Add $x$ to $\mathcal{C}$.

Note that a cover with radius $r$ satisfies the following two properties: every point in $X$ is within distance $r$ to some point in $\mathcal{C}$ (under the same metric $d'$), and all points in $\mathcal{C}$ are at least distance $r$ apart from each other.

We now introduce the cover tree datastructure of Beygelzimer et al. (2006). For the data structure, we create a sequence of covers $\mathcal{C}_{-1}, \mathcal{C}_0, \ldots$. Every $\mathcal{C}_i$ is a layer in the final Cover Tree $\mathcal{T}$.

---

**Algorithm 2** Cover Tree Data structure

---

1: **Input:** A set $X$ of $n$ points, metric $d$, real number $T \geq 1$.
2: **Output:** A tree on $X$
3: **procedure** COVER-TREE$(d, T)$
4:     WLOG, all distances between points in $X$ under $d$ are in $(1, \Delta]$ by scaling.
5:     $\mathcal{C}_{-1} = \mathcal{C}_0 = X$
6:     $\mathcal{C}_i$ is a $2^i/T$-cover of $\mathcal{C}_{i-1}$ for $i > 0$ under metric $d$
7:     $\mathcal{C}_i \subseteq \mathcal{C}_{i-1}$ for $i > 0$.
8:     $t = O(\log(\Delta T))$                               $\triangleright$ $t$ is the number of levels of $\mathcal{T}$
9:     **for** $i = -1$ to $t$ **do**
10:         $\mathcal{C}_i$ corresponds to tree nodes of $\mathcal{T}$ on level $i$
11:         Each $p \in \mathcal{C}_{i-1} \setminus \mathcal{C}_i$ is connected to exactly one $p \in \mathcal{C}_i$ such that $d(p, p') \leq 2^i/T$
12:     **Return** tree $\mathcal{T}$

---

**Lemma C.2** (Theorem 1 in Beygelzimer et al. (2006)). *$\mathcal{T}$ takes $O(n)$ space, regardless of the value of $T$.*

*Proof.* We use the *explicit* representation of $\mathcal{T}$ (as done in Beygelzimer et al. (2006)), where we coalesce all nodes in which the only child is a self-child. Thus, every node either has a parent other than the self-parent or a child other than the self-child. This gives an $O(n)$ space bound, independent of all other parameters. $\qquad\square$

We note that it is possible to construct the cover tree data structure of Algorithm 2 in time $2^{O(\lambda_d)} n \log n$, but it is not important to our discussion Beygelzimer et al. (2006).

Now we describe the query procedure. Here, we can query with a metric $D$ that is possibly different than the metric $d$ used to create $\mathcal{T}$ in Algorithm 2.

---

**Algorithm 3** Cover Tree Search

---

1: **Input:** Cover tree $\mathcal{T}$ associated with point set $X$, query point $q$, metric $D$, accuracy $\varepsilon \in (0, 1)$.
2: **Output:** A point $p \in X$
3: **procedure** COVER-TREE-SEARCH
4:     $t \leftarrow$ number of levels of $\mathcal{T}$
5:     $Q_t \leftarrow \mathcal{C}_t$                                    $\triangleright$ We use the covers that define $\mathcal{T}$
6:     $i \leftarrow t$
7:     **while** $i \neq -1$ **do**
8:         $Q = \{p \in \mathcal{C}_{i-1} : p \text{ has a parent in } Q_i\}$
9:         $Q_{i-1} = \{p \in Q : D(q, p) \leq D(q, Q) + 2^i\}$
10:         **if** $D(q, Q_{i-1}) \geq 2^i(1 + 1/\varepsilon)$ **then**
11:             Exit the while loop.
12:         $i \leftarrow i - 1$
13:     **Return** point $p \in Q_{i-1}$ that is closest to $q$ under $D$

---

### C.0.2   THE MAIN THEOREM

We construct a cover tree $\mathcal{T}$ using metric $d$ and $T$ from Equation 1 in Algorithm 2. Upon a query $q$, we search for an approximate nearest neighbor in $\mathcal{T}$ in Algorithm 3, using metric $D$ instead. Our main theorem is the following.

**Theorem C.3.** *Let $Q_{\mathtt{CoverTree}}(\varepsilon, \Delta_d, \lambda_d) = 2^{O(\lambda_d)} \log(\Delta_d) + (1/\varepsilon)^{O(\lambda_d)}$ denote the query complexity of the standard cover tree datastructure, where we set $T = 1$ in Algorithm 2 and build and search using the same metric $d$. Now consider two metrics $d$ and $D$ satisfying Equation 1. Suppose we build a cover tree $\mathcal{T}$ with metric $d$ by setting $T = C$ in Algorithm 2, but search using metric $D$ in Algorithm 3. Then the following holds:*

    *1. The space used by $\mathcal{T}$ is $O(n)$.*

---

2. *Running Algorithm 3 using $D$ finds a $1 + \varepsilon$ approximate nearest neighbor of $q$ in the dataset $X$ (under metric $D$).*

3. *On any query, Algorithm 3 invokes $D$ at most*

$$C^{O(\lambda_d)} \log(\Delta_d) + (C/\varepsilon)^{O(\lambda_d)} = \tilde{O}(Q_{\texttt{CoverTree}}(\Omega(\varepsilon/C), \Delta_d, \lambda_d)).$$

*times.*

Two prove Theorem C.3, we need to: (a) argue correctness and (b) bound the number of times Algorithm 3 calls its input metric $D$. While both follow from similar analysis as in Beygelzimer et al. (2006), it is not in a black-box manner since the metric we used to search $\mathcal{T}$ in Algorithm 3 is different than the metric used to build $\mathcal{T}$ in Algorithm 2.

We begin with a helpful lemma.

**Lemma C.4.** *For any $p \in \mathcal{C}_{i-1}$, the distance between $p$ and any of its descendants in $\mathcal{T}$ is bounded by $2^i$ under $D$.*

*Proof.* The proof of the lemma follows from Theorem 2 in Beygelzimer et al. (2006). There, it is shown that for any $p \in \mathcal{C}_{i-1}$ the distance between $p$ and any descendant $p'$ is bounded by $d(p, p') \leq \sum_{j=-\infty}^{i-1} 2^j / T = 2^i / T$, implying the lemma after we scale by $C$ due to Equation 1 (note we set $T = C$ in the construction of $\mathcal{T}$ in Theorem C.3). $\square$

We now argue accuracy.

**Theorem C.5.** *Algorithm 3 returns a $1 + \varepsilon$-approximate nearest neighbor to query $q$ under $D$.*

*Proof.* Let $p^*$ be the true nearest neighbor of query $q$. Consider the leaf to root path starting from $p^*$. We first claim that if $Q_i$ contains an ancestor of $p^*$, then $Q_{i-1}$ also contains an ancestor $q_{i-1}$ of $p^*$. To show this, note that $D(p^*, q_{i-1}) \leq 2^i$ by Lemma C.4, so we always have

$$D(q, q_{i-1}) \leq D(q, p^*) + D(p^*, q_{i-1}) \leq D(q, Q) + 2^i,$$

meaning $q_{i-1}$ is included in $Q_{i-1}$.

When we terminate, either we end on a single node, in which case we return $p^*$ exactly (from the above argument), or when $D(q, Q_{i-1}) \geq 2^i(1 + 1/\varepsilon)$. In this latter case, we additionally know that

$$D(q, Q_{i-1}) \leq D(q, p^*) + D(p^*, Q_{i-1}) \leq D(q, p^*) + 2^i$$

since an ancestor of $p^*$ is contained in $Q_{i-1}$ (namely $q_{i-1}$ from above). But the exit condition implies

$$2^i(1 + 1/\varepsilon) \leq D(q, p^*) + 2^i \implies 2^i \leq \varepsilon D(q, p^*),$$

which means

$$D(q, Q_{i-1}) \leq D(q, p^*) + 2^i \leq D(q, p^*) + \varepsilon D(q, p^*) = (1 + \varepsilon)D(q, p^*),$$

as desired. $\square$

Finally, we bound the query complexity. The following follows from the arguments in Beygelzimer et al. (2006).

**Theorem C.6.** *The number of calls to $D$ in Algorithm 3 is bounded by $C^{O(\lambda_d)} \cdot \log(\Delta_d C) + (C/\varepsilon)^{O(\lambda_d)}$.*

*Proof Sketch.* The bound follows from Beygelzimer et al. (2006) but we briefly outline it here. The query complexity is dominated by the size of the sets $Q_{i-1}$ in Line 9 as the algorithm proceeds. We give two ways to bound $Q_{i-1}$. Before that, note that the points $p$ that make up $Q_{i-1}$ are in a cover (under $d$) by the construction of $\mathcal{T}$, so they are all separated by distance at least $\Omega(2^i/C)$ (under $d$). Let $p^*$ be the closest point to $q$ in $X$.

- **Bound 1**: In the iterations where $D(q, p^*) \leq O(2^i)$, we have the diameter of $Q_{i-1}$ under $D$ is at most $O(2^i)$ as well. This is because an ancestor $q_{i-1} \in C_{i-1}$ of $p^*$ is in $Q$ of line 8 (see proof of Theorem C.5), meaning $D(q, Q) \leq O(2^i)$ due to Lemma C.4. Thus, any point $p \in Q_{i-1}$ satisfies $D(q, p) \leq D(q, Q) + 2^i = O(2^i)$. From Equation 1, it follows that the diameter of $Q_{i-1}$ under $d$ is also at most $O(2^i)$. We know the points in $Q_{i-1}$ are separated by mutual distance at least $\Omega(2^i/C)$ under $d$, implying that $|Q_{i-1}| \leq C^{O(\lambda_d)}$ in this case by a standard packing argument. This case can occur at most $O(\log(\Delta C))$ times, since that is the number of different levels of $\mathcal{T}$.

- **Bound 2**: Now consider the case where $D(q, p^*) \geq \Omega(2^i)$. In this case, we have that the points in $Q_{i-1}$ have diameter at most $O(2^i/\varepsilon)$ from $q$ (under $D$), due to the condition of line 10. Thus, the diameter is also bounded by $O(2^i/\varepsilon)$ under $d$. By a standard packing argument, this means that $|Q_{i-1}| \leq (C/\varepsilon)^{O(\lambda_d)}$, since again $Q_{i-1}$ are mutually separated by distance at least $\Omega(2^i/C)$ under $d$. However, our goal is to show that the number of iterations where this bound is relevant is at most $O(\log(1/\varepsilon))$. Indeed, we have $D(q, Q_{i-1}) \leq O(2^i/\varepsilon)$, meaning $2^i \geq \Omega(\varepsilon D(q, Q_{i-1})) \geq \Omega(\varepsilon D(q, p^*))$ Since we are decrementing the index $i$ and are in the case where $D(q, p^*) \geq \Omega(2^i)$, this can only happen for $O(\log(1/\varepsilon))$ different $i$'s.

Combining the two bounds proves the theorem. $\qquad\square$

The proof of Theorem C.3 follows from combining Lemmas C.2 and Theorems C.5 and C.6.

# D    COMPLETE EXPERIMENTAL RESULTS

**Parameter choices for Nearest Neighbor Search algorithms**    The parameter choices for DiskANN are $\alpha = 1.2$, $l\_build = 125$, $max\_outdegree = 64$ (the standard choices used in ANN benchmarks Aumüller et al. (2020)). The parameter choices for NSG are the same as the authors' choices for GIST1M dataset (Jégou et al., 2011): $K = 400$, $L = 400$, $iter = 12$, $S = 15$, $R = 100$. NSG also requires building a knn-graph with efanna, where we use the standard parameters: $L = 60$, $R = 70$, $C = 500$.

**Empirical Results**    We report the empirical results of using different embedding models as distance proxy, using the NSG algorithm, and measuring Recall@10.

1. We report the results of using "bge-micro-v2" as the distance proxy $d$ and using DiskANN for building the graph index. See Figure 4 for Recall@10 metric plots.

2. We report the results of using "gte-small" as the distance proxy $d$ and using DiskANN for building the graph index. See Figure 5 for NDCG@10 metric plots and Figure 6 for Recall@10 metric plots.

3. We report the results of using "bge-base-en-v1,5" as the distance proxy $d$ and using DiskANN for building the graph index. See Figure 7 for NDCG@10 metric plots and Figure 8 for Recall@10 metric plots.

4. We report the results of using "bge-micro-v2" as the distance proxy $d$ and using NSG for building the graph index. See Figures 9 for NDCG@10 metric plots and  10 for Recall@10 metric plots.

We can see that for all the different cheap distance proxies ("bge-micro-v2" Xiao et al. (2023), "gte-small" Li et al. (2023), "bge-base-en-v1.5" Xiao et al. (2023)) and both nearest neighbor search algorithms (DiskANN Jayaram Subramanya et al. (2019) and NSG Fu et al. (2019b)), our method has better NDCG and Recall results on most datasets. Moreover, naturally the advantage of our method over Bi-metric (baseline) is larger when there is a large gap between the qualities of the cheap distance proxy $d$ and the ground truth distance metric $D$. This makes sense because as their qualities converge, the cheap proxy alone is enough to retrieve the closest points to a query for the expensive metric $D$.

We also report the histograms of empirical $C = d/D$ values using "bge-micro-v2' as the distance proxy $d$ in Figure 11. For all 15 datasets, the distance ratio $C = d/D$ concentrates well around 1

**Different nearest neighbor search algorithms**   We implement our method with another popular empirical nearest neighbor search algorithm called NSG Fu et al. (2019b). We obtain the same qualitative behavior as DiskANN. Because the authors' implementation of NSG only supports $\ell_2$ distances, we first normalize all the embeddings and search via $\ell_2$. This may cause some performance drops. Therefore, we are not comparing the results between the DiskANN and NSG algorithms, but only results from different methods, fixing the graph index. In Figure 9 and 10 in the appendix, we observe that our method still performs the best compared to Bi-metric (baseline) and single metric in most cases, demonstrating that our bi-metric framework can be applied to other graph-based nearest neighbor search algorithms as well.

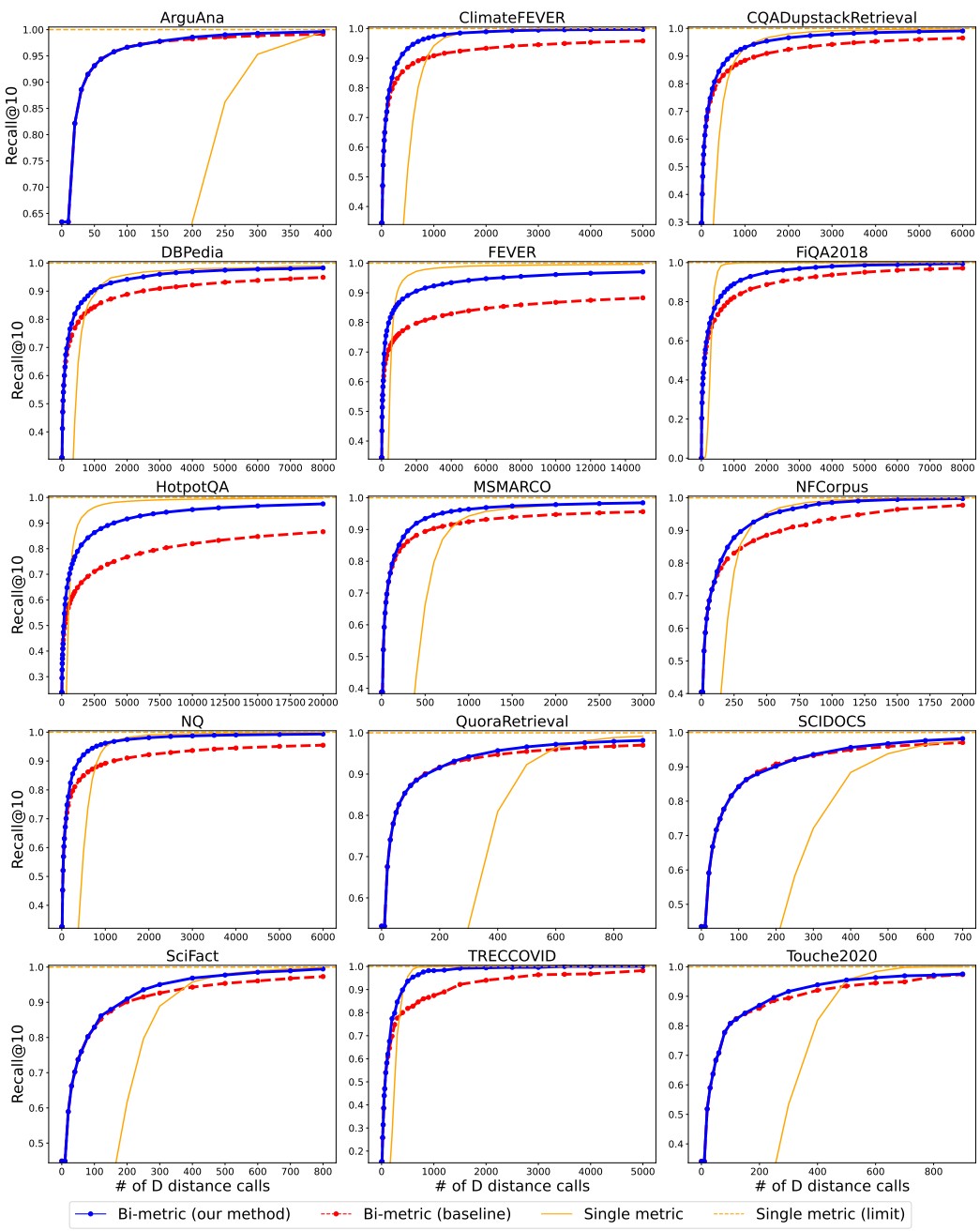

Figure 4: Results for 15 MTEB Retrieval datasets. The x-axis is the number of expensive distance function calls. The y-axis is the Recall@10 score. The cheap model is "bge-micro-v2", the expensive model is "SFR-Embedding-Mistral", and the nearest neighbor search algorithm used is DiskANN.

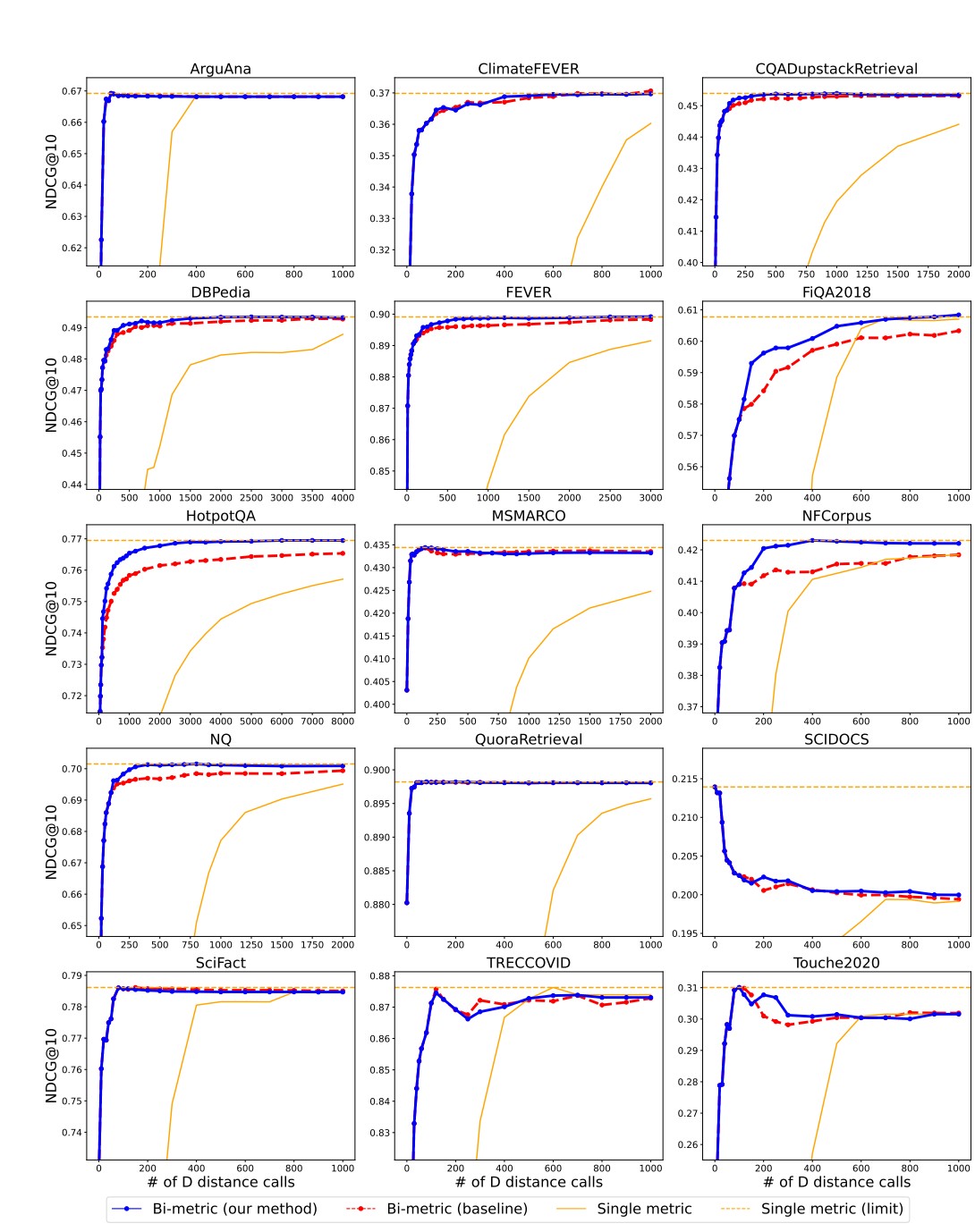

Figure 5: Results for 15 MTEB Retrieval datasets. The x-axis is the number of expensive distance function calls. The y-axis is the NDCG@10 score. The cheap model is "gte-small", the expensive model is "SFR-Embedding-Mistral", and the nearest neighbor search algorithm used is DiskANN.

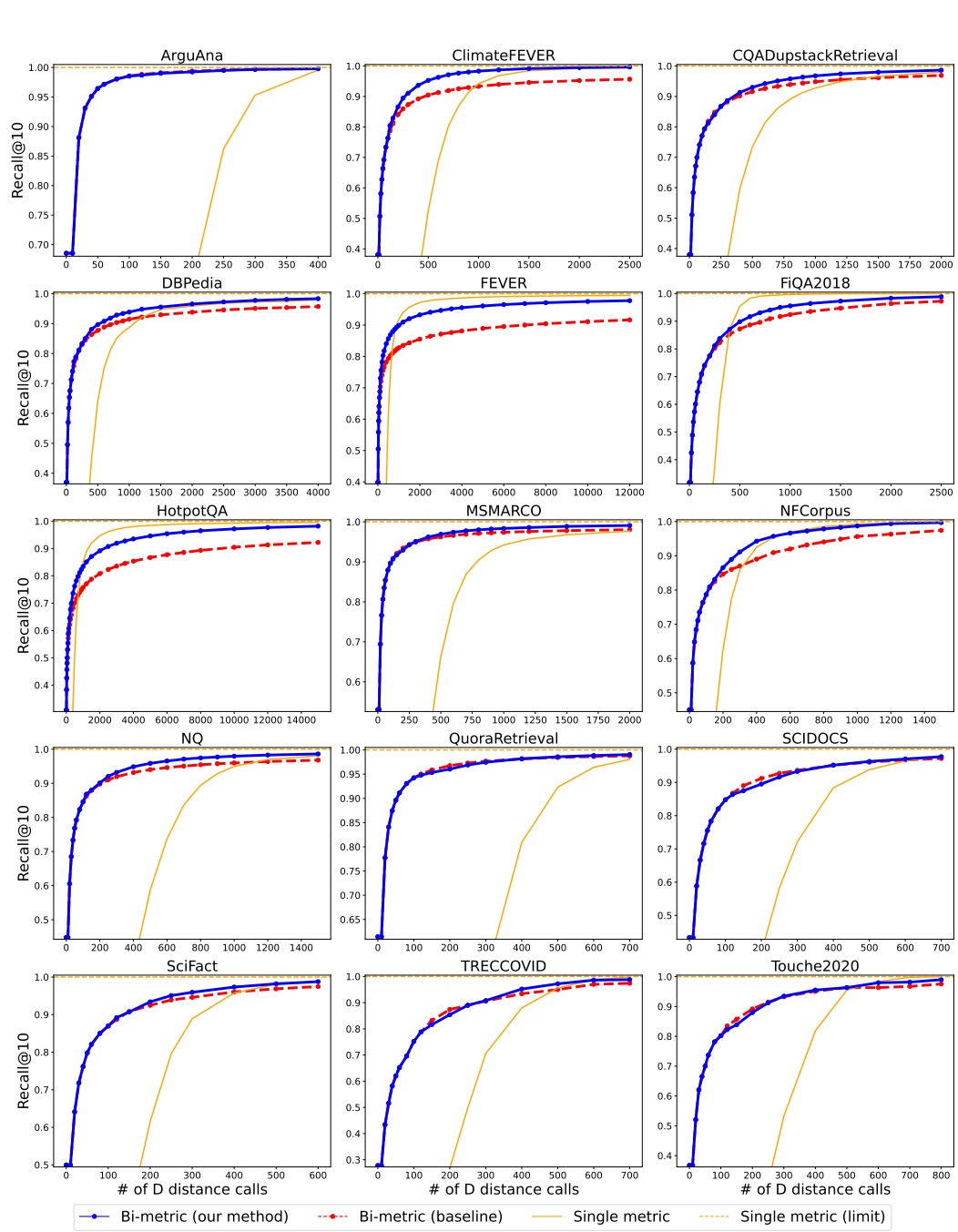

Figure 6: Results for 15 MTEB Retrieval datasets. The x-axis is the number of expensive distance function calls. The y-axis is the Recall@10 score. The cheap model is "gte-small", the expensive model is "SFR-Embedding-Mistral", and the nearest neighbor search algorithm used is DiskANN.

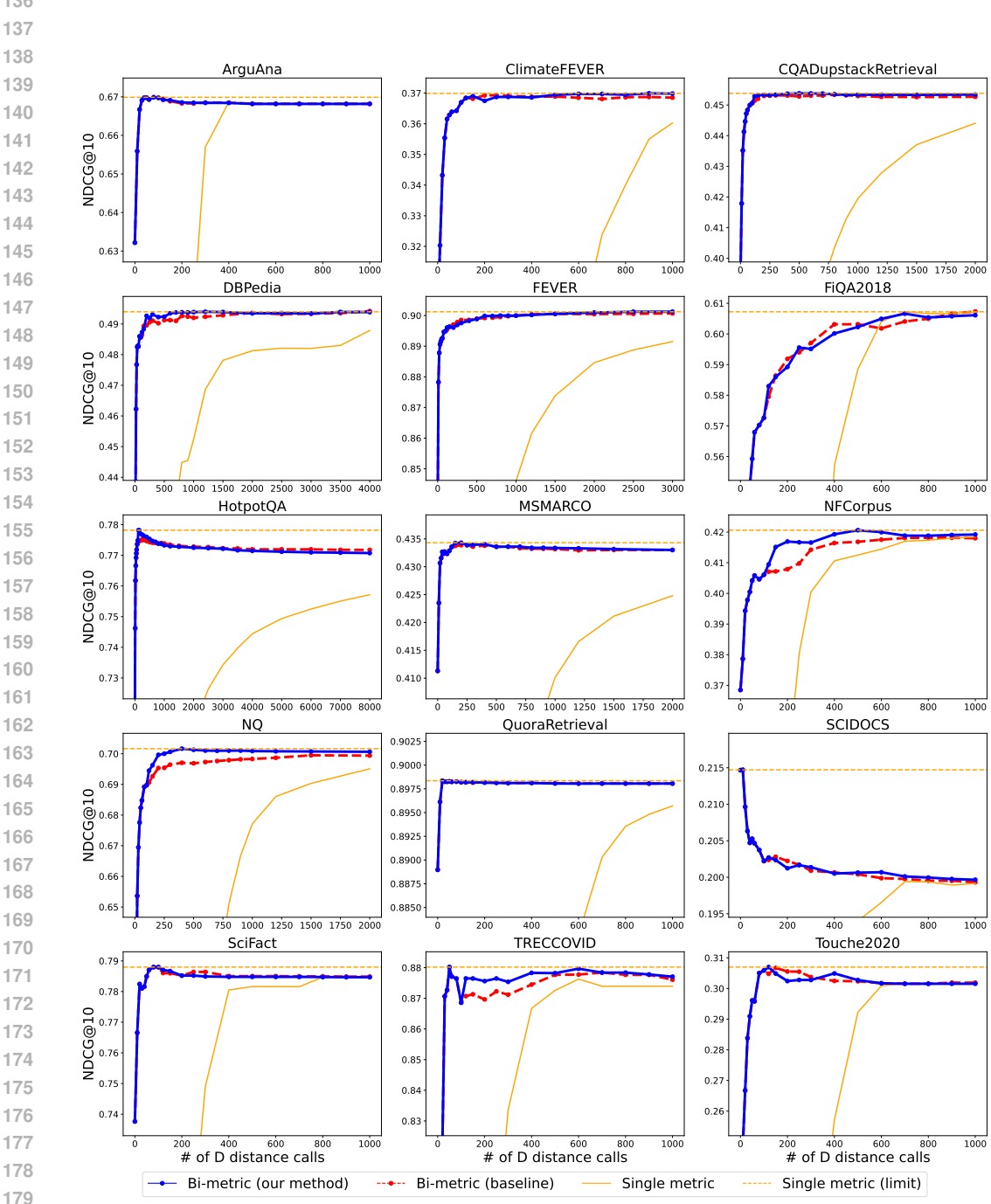

Figure 7: Results for 15 MTEB Retrieval datasets. The x-axis is the number of expensive distance function calls. The y-axis is the NDCG@10 score. The cheap model is "bge-base-en-v1.5", the expensive model is "SFR-Embedding-Mistral", and the nearest neighbor search algorithm used is DiskANN.

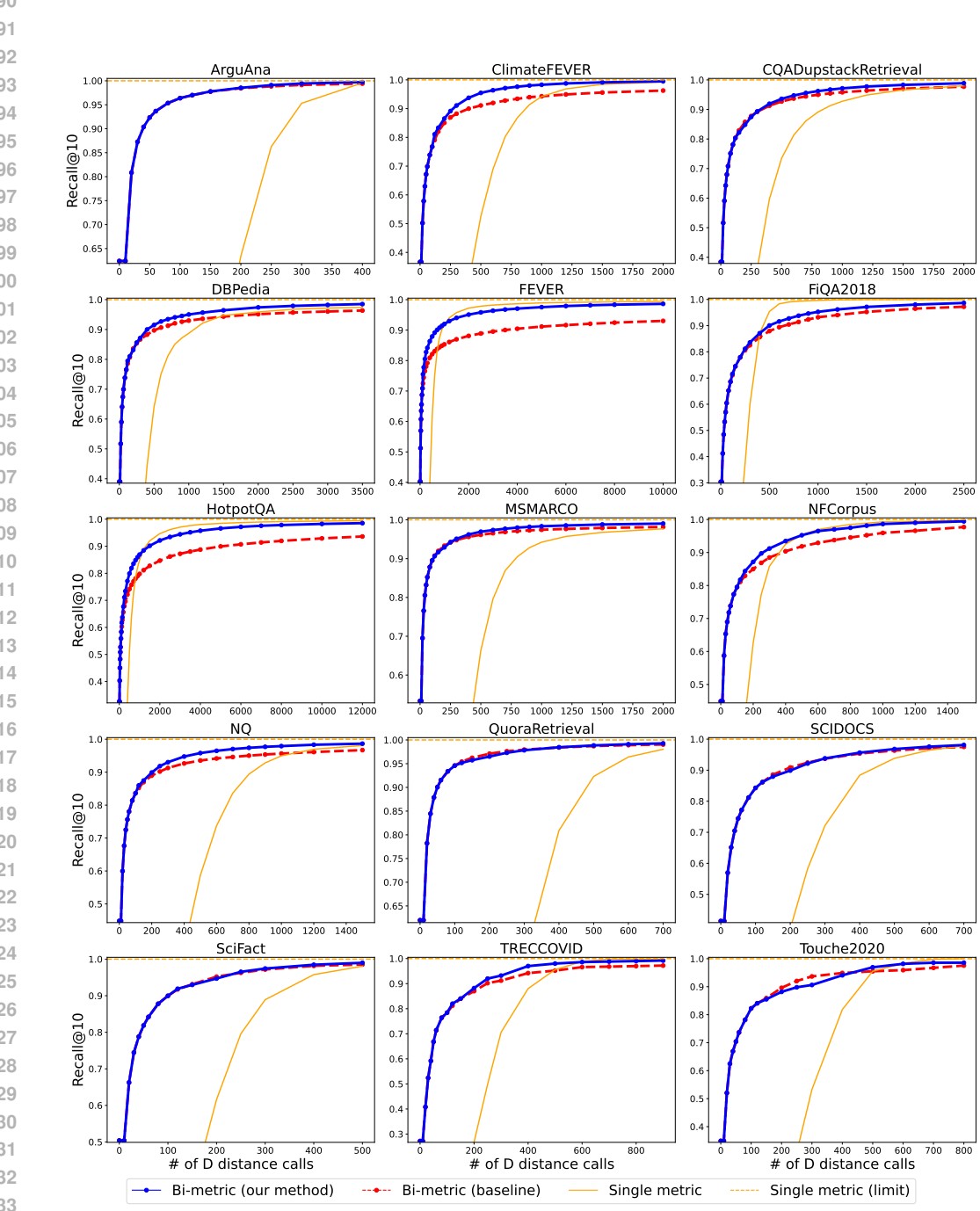

Figure 8: Results for 15 MTEB Retrieval datasets. The x-axis is the number of expensive distance function calls. The y-axis is the Recall@10 score. The cheap model is "bge-base-en-v1.5", the expensive model is "SFR-Embedding-Mistral", and the nearest neighbor search algorithm used is DiskANN.

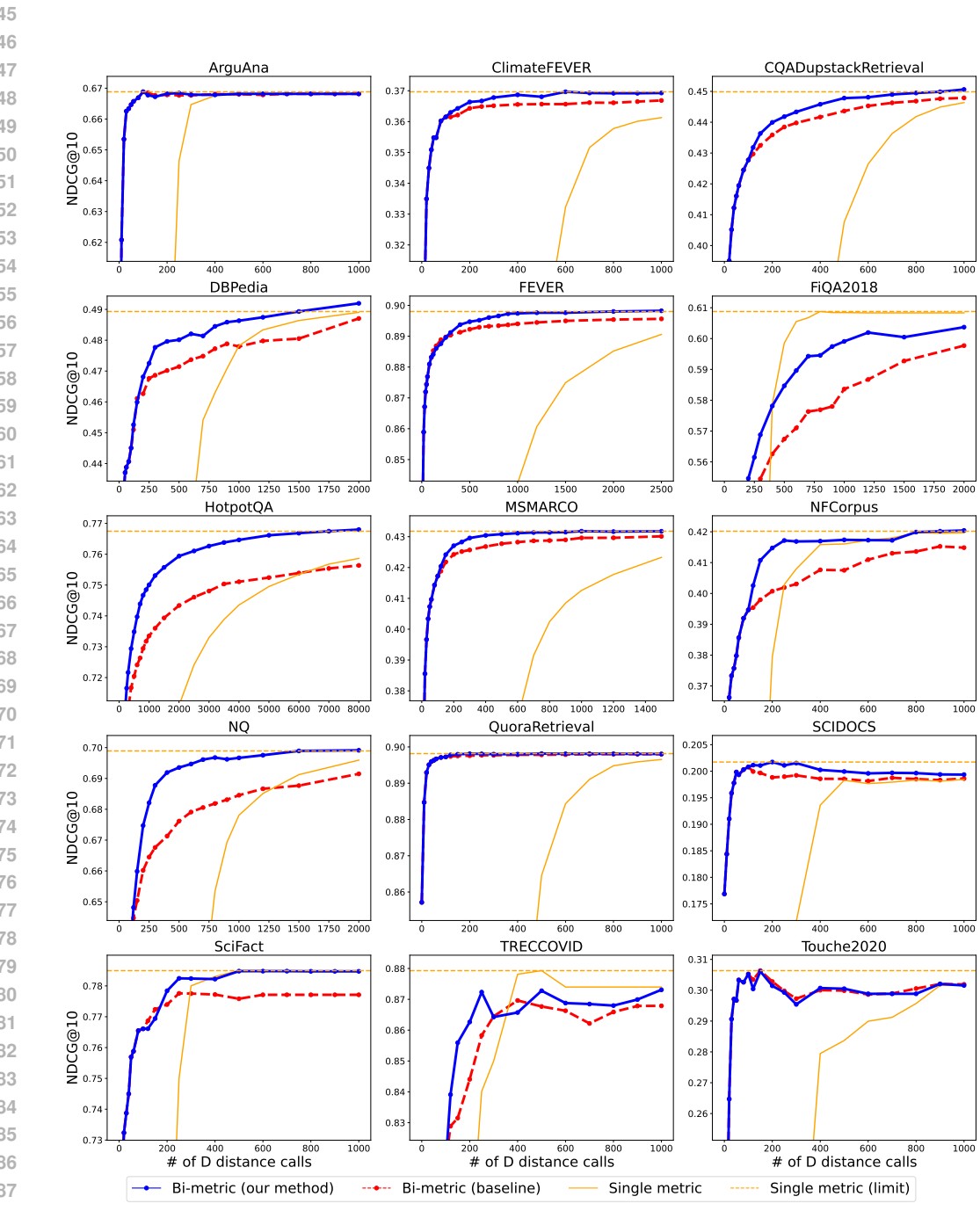

Figure 9: Results for 15 MTEB Retrieval datasets. The x-axis is the number of expensive distance function calls. The y-axis is the NDCG@10 score. The cheap model is "bge-micro-v2", the expensive model is "SFR-Embedding-Mistral", and the nearest neighbor search algorithm used is NSG.

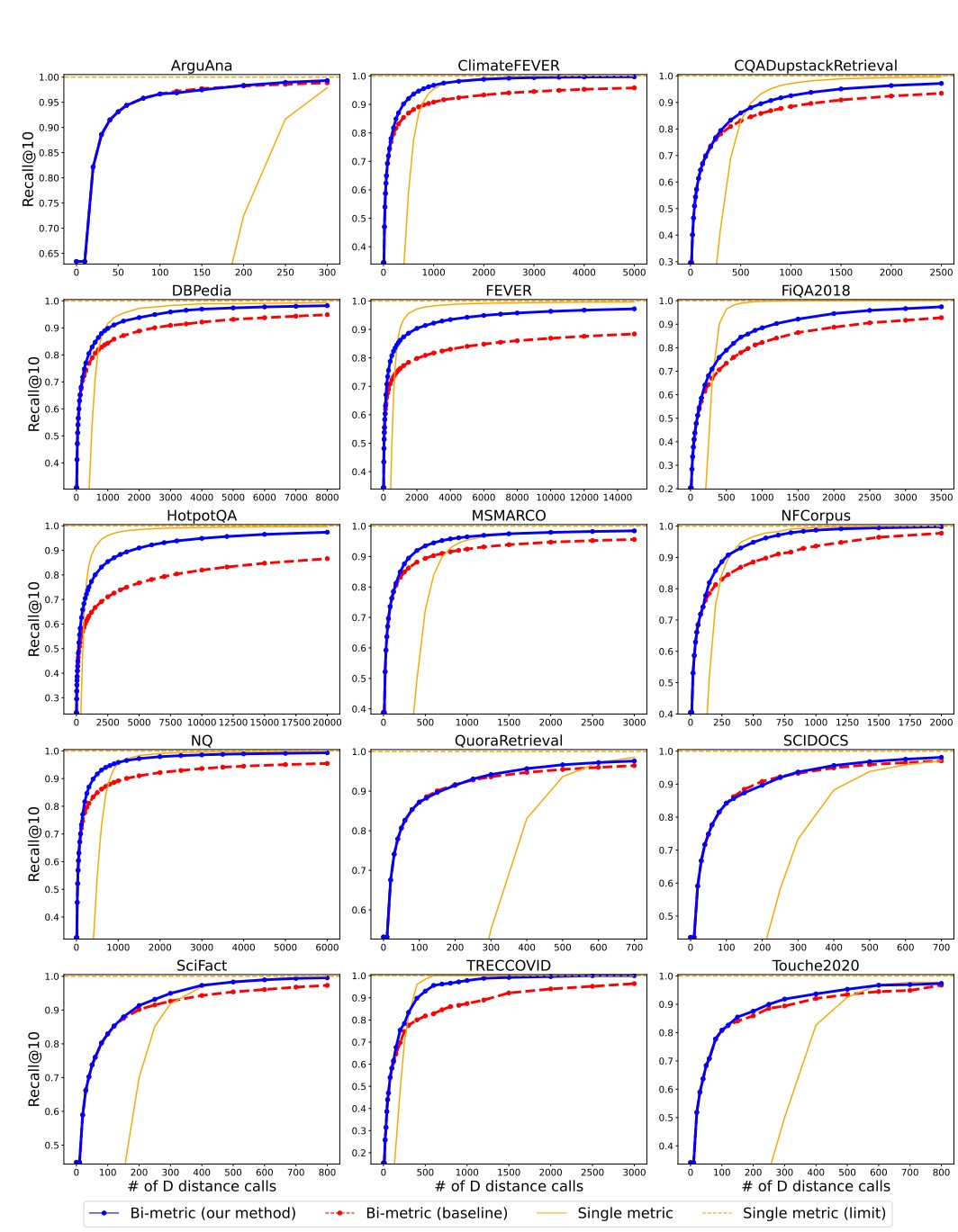

Figure 10: Results for 15 MTEB Retrieval datasets. The x-axis is the number of expensive distance function calls. The y-axis is the Recall@10 score. The cheap model is "bge-micro-v2", the expensive model is "SFR-Embedding-Mistral", and the nearest neighbor search algorithm used is NSG.

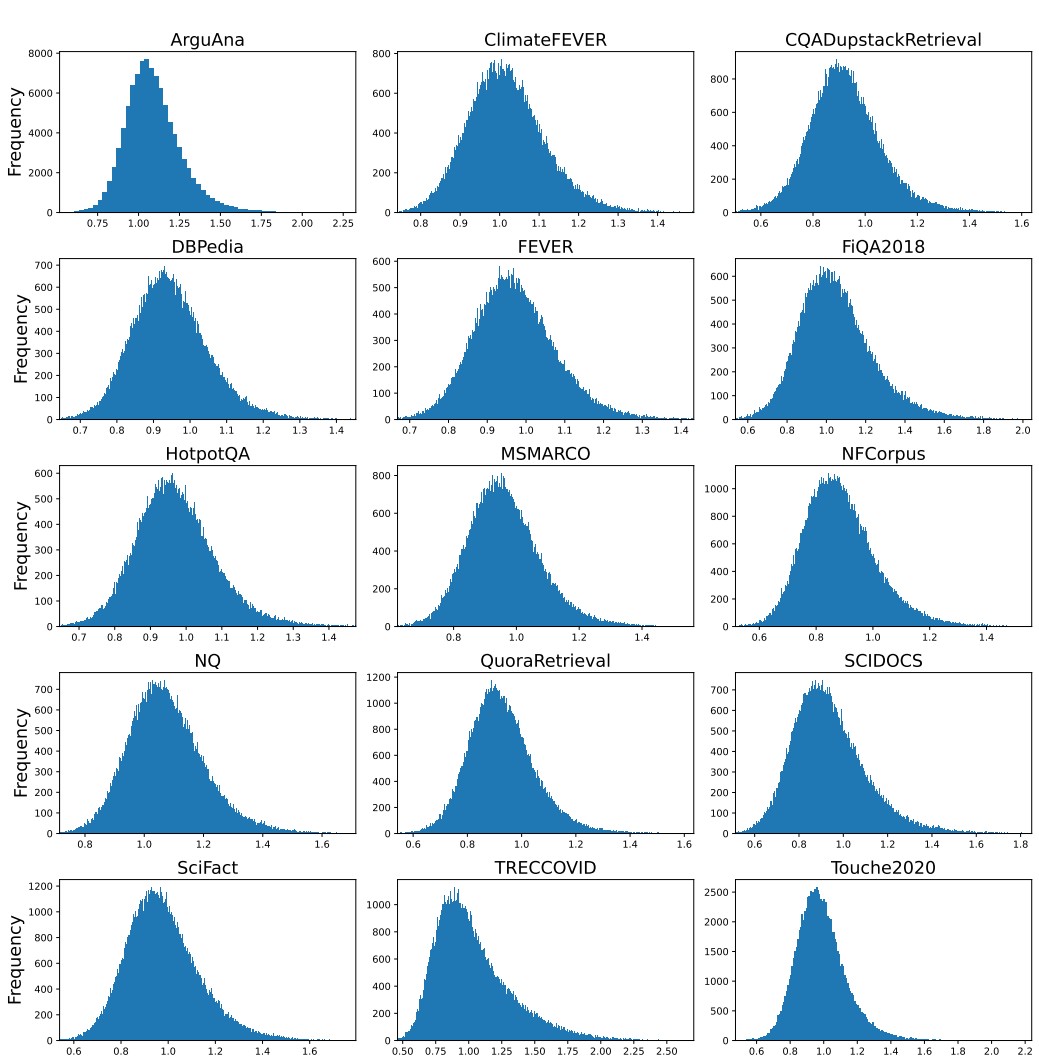

Figure 11: Results for 15 MTEB Retrieval datasets. Histograms of $C = D/d$ values, where we use "bge-micro-v2" as the distance proxy $d$ and "SFR-Embedding-Mistral" as the expensive distance $D$.

