# OpenReview forum: "A Bi-metric Framework for Efficient Nearest Neighbor Search"
_ICLR.cc/2025/Conference — ICLR 2025 Conference Withdrawn Submission_

### Official Review · Reviewer_qYPh · 2024-11-03

**Soundness:** 2
**Presentation:** 2
**Contribution:** 3
**Rating:** 3
**Confidence:** 3

**Summary:**

This paper presents a general bi-metric framework for efficient nearest neighbor search by combining an expensive, accurate ground-truth metric $D$ and a cheap, approximate proxy metric $d$. The framework constructs data structures based on the proxy metric $d$ for efficiency, while using limited calls to metric $D$ in query and retaining the accuracy of the ground-truth metric $D$. The authors demonstrate this framework with DiskANN and Cover Tree algorithms and apply it to text retrieval tasks, indicating improvements in the accuracy-efficiency trade-off compared to existing re-ranking or bi-metric methods.

**Strengths:**

1. Theoretical guarantees of the approximation quality (accuracy) are provided.

2. Although methods that combine a cheap metric $d$ with an expensive metric $D$ already exist, the proposed method addresses the limitations of existing methods: (1) The proposed method achieves the accuracy of the expensive metric $D$; (2) The proposed method requires only a sub-linear number of evaluations of $d$ and $D$ during the query stage. These two points represent significant improvements in the nearest neighbor search problem.

3. Extensive experiments across 15 MTEB retrieval datasets were conducted and showed consistent performance gains.

**Weaknesses:**

1. The proposed framework is applicable only to algorithms that use the concept of an $r$-net (where, given a parameter $r$, an $r$-net is a small subset of the dataset that ensures every data point is within distance $r$ to the subset in the net).
For nearest neighbor algorithms that do not rely on $r$-net, the proposed framework is likely inapplicable.

2. The proposed framework lacks a clear explanation, making it difficult to assess the novelty of the algorithm.
The paper does not provide pseudocode in the main text or use toy examples to help readers understand the algorithm.
The proposed algorithm is only partially described in Section 4, while Section 3 mainly focuses on theoretical analysis, leaving readers somewhat lost, as they do not fully grasp the proposed method at this point.

3. The experiment results show only the number of calls to metric $D$, while the number of calls to metric $d$ is not shown.
Since a key claimed improvement over existing bi-metric methods is the reduction (or even sub-linear scaling) in calls to metric $d$, it would be valuable to report the number of calls to $d$ as well.

4. It would be valuable to evaluate across datasets with varying intrinsic dimensionalities.
Since the bi-metric framework depends on the proxy metric $d$ closely approximating the ground-truth metric $D$, datasets with higher intrinsic dimensionality may introduce greater approximation error, making it more challenging to achieve accurate results.
Additionally, higher intrinsic dimensionality could increase query complexity and reduce the efficacy of the $r$-net structure used in the method, potentially impacting both accuracy and efficiency.
Testing the framework across datasets with different intrinsic dimensionalities would clarify its robustness and scalability under diverse data conditions.

5. In several places, such as line 98 (Equation 1) and line 279 (Algorithm 1), the main text references equations or algorithms that appear much later or only in the appendix.
This disrupts the continuity and can cause confusion for readers.
It would be helpful if the authors could either move these elements earlier in the text or provide clearer guidance to improve readability and flow.

**Questions:**

1. Could you please discuss how your proposed framework applied to other commonly used data structures like Locality-Sensitive Hashing, Proximity Graph, K-d Tree, and Product Quantization? (W1)

2. Could you consider reorganizing the content (including pseudocode in the main text) to improve clarity and provide a more cohesive flow? (W2)

3. Moreover, could you provide a step-by-step explanation of the proposed algorithm, possibly with a toy example, to enhance clarity and help readers fully understand the method before delving into the theoretical analysis? (W2)

4. The paper states that only a sub-linear number of evaluations of both $d$ and $D$ are required during querying. However, it seems that the sub-linear evaluations of $d$ are neither illustrated nor proven, and this is not immediately clear. How does the proposed algorithm achieve a sub-linear number of distance computations? (W3)

5. How does the proposed framework handle datasets with different intrinsic dimensionalities, particularly regarding the accuracy of the proxy metric and the effectiveness of the $r$-net structure under these conditions? (W4)

6. Could the authors consider restructuring the placement of key equations and algorithms, or provide additional guidance to improve readability and continuity throughout the paper? (W5)

---

> ### Author Response · Authors · 2024-11-22
> **Response to Reviewer qYPh**
>
> Thank you for your feedback. We hope that we can clarify a few points.
>
> > Weakness 1
>
> We instantiate our framework on both DiskANN and CoverTree (in both theory and empirically) and on NSG (empirically, as we are not aware of any theoretical analysis of this algorithm). We believe this is sufficient to demonstrate the applicability of our framework.  DiskANN is among the best empirical algorithms in practice, as per https://ann-benchmarks.com. In our experiments, we are able to substantially improve over the widely used “re-ranking” benchmark and achieve state-of-the-art retrieval accuracy using up to 4x fewer evaluations of the expensive model. We also show consistent performance boosts across a wide collection of 15 datasets in MTEB.
> We also discuss other techniques in the paragraph starting from line 254 and the introduction.
>
> > Weakness 2
>
> Our framework is discussed in detail in line 187 and its instantiation to DiskANN and Covertree are given in full details in Algorithms 1 and 2 in the appendix.
>
> > Weakness 3
>
> The total runtime spent evaluating d, the cheap metric, for our datasets is insignificant compared to calling D, the expensive metric. This is discussed extensively in our paper. For example in line 395, we note that the *total* time spent evaluating d is 0.37s per query for our largest dataset, HotpotQA. However, *each* evaluation to D takes 0.13s. Since us (and baseline methods) need to make > 2000 queries to D to get decent accuracy, the cost to evaluate D is > 500x more costly. Thus, we believe it is justified to only plot the evaluations to D in our main figures.
>
> > Weakness 4
>
> While we explicitly don't compute the intrinsic dimensionality of our datasets, we show the power of our method via consistent performance boosts across a wide collection of 15 datasets in MTEB. We believe this is sufficient evidence for both applicability and scalability of our method.
>
>
> > Weakness 5
>
> Thank you for the feedback. We will clarify these points in the next version.
>
> > Question 1
>
> Please see section 3 which discusses the reasons why our methods work for the graph-based indices, and not for algorithms which are substantially different such as locality-sensitive hashing. We note that graph based indices are already among the most popular choice of algorithms in practice and our strong empirical results corroborate this fact.
>
> > Question 2
>
> Thank you, we will consider your comments in the next version of the paper.
>
> > Question 3
>
> Our framework is explained in detail starting from line 187, with intuition given about why it is applicable to graph based nearest neighbor data structure in the 4 paragraphs in Section 3.
>
> > Question 4
>
> The sublinear bound is proven in Theorem 1.1. Our query times have only a logarithmic dependence on n, the size of our dataset.
>
> > Question 5
>
> The theoretic dependence on the doubling dimension is given our our formal theorem statements. Empirically, our method obtains the best or matches prior baseline, obtaining up to 4x reduction in the number of expensive evaluations across a wide array of 15 MTEB datasets.
>
> > Question 6
>
> Thank you for the feedback about organization. We will consider these points in the future version.

---

> > ### Comment · Reviewer_qYPh · 2024-11-24
> >
> > Thank you for your detailed response! It partially addressed my concerns, but I have some follow-up comments and questions:
> >
> > **Regarding (W2), (W5), and (Q3):**
> >
> > Thank you for your response. Line 187 provides a general description of the framework but lacks detailed explanations. It would be helpful to delve deeper into how the framework can specifically enhance the performance of DiskANN and CoverTree.
> >
> > Moreover, since Algorithms 1 and 2 are frequently referenced in the main text, it would be beneficial to move them into the main text and refine the flow of presenting these specific applications of the general framework to the respective data structures. Currently, the structure feels somewhat unclear, which can make it difficult for readers to follow.
> >
> > I trust that the next version will feature an improved presentation and a clearer structure for the methods and equations.
> >
> > **Regarding (W3) and (Q4):**
> >
> > I agree that the total evaluation costs of $D$ can be more expensive than those of $d$. However, I would appreciate it if you could provide additional results on the costs of evaluating $d$ and explain why these costs are considered insignificant.
> >
> > Additionally, after revisiting Theorem 1.1, I was unable to find a proof of sub-linear evaluations for $d$ (only proof for $D$ found). Could you provide direct proof or further explanation regarding this?
> >
> > **Regarding (W4) and (Q5):**
> >
> > My question initially aimed to understand whether the effectiveness of the proposed method is influenced by variations in intrinsic dimensionality and whether the method relies on the assumption that $d$ provides accurate approximations of $D$ (i.e., $d(x, y) \leq D(x, y) \leq C \cdot d(x, y)$, this could be a challenging requirement to fulfill in real-world since we may not have such a specific value of $d$)?
> >
> > I would appreciate it if you could provide further insights or additional results on the impact of varying intrinsic dimensions or varying $C$ values on the method's performance.

---

> ### Author Response · Authors · 2024-11-24
>
> Thank you for your response! We would like to provide additional explanations in the following:
>
> **Regarding (W2), (W5), and (Q3):**
>
> We admit that the description of our framework is somewhat vague, and we will clarify it in a later version. However, we want to emphasize that vagueness and generalizability are natural differences between "a framework" and "a method." We are proposing a new framework to solve the nearest neighbor search problem, within which we can apply many existing algorithms (such as DiskANN and CoverTree). Please refer to our framework descriptions in lines 70-90 and 187-199, and the detailed method description in lines 317-339 in the experiment section.
>
> Both our theoretical and empirical results support the following points:
>
> This framework is effective, showing significant improvements in reducing the number of expensive distance computations across different experimental settings. (see Figures 1, 4, 5, 6, 7, 8, 9, and 10).
>
> This framework is generalizable: we tested 4 different embedding models (see Table 1), on 15 datasets, and with 3 nearest neighbor search algorithms (DiskANN, NSG, CoverTree), providing both empirical results and theoretical analysis.
> Finally, thank you for your suggestions on our presentation!
>
>
> **Regarding (W3) and (Q4):**
>
> For the cost of using d and D in query answering, please refer to our response to your Weakness 3 and our experimental analysis from lines 395-405.
>
> For the cost of solely evaluating sentence embeddings using d and D, please see the detailed computational resource comparison from lines 201-206.
>
> Additionally, please refer to lines 200–215 for a list of reasons (e.g., time, disk storage, API cost, training flexibility) why using a combination of d and D has advantages over classical methods that rely only on d or D.
>
> Regarding sublinear evaluations for d: Theorem 1.1 is a general statement for our framework. For its instantiation with DiskANN, we suggest that readers also look at Theorem 3.4.
> In short, our framework "builds the index using d and searches for the query using D." Therefore, d is only involved in the indexing phase (not the query phase), and the sublinear number of d calls follows the standard analysis for DiskANN or CoverTree with metric d. Please refer to the DiskANN analysis in Indyk & Xu (2023) and the CoverTree analysis in Beygelzimer et al. (2006).
>
> **Regarding (W4) and (Q5):**
>
> We argue that the intrinsic dimensionality (e.g., doubling dimension) of a dataset is more likely to be a theoretical definition adopted in the literature, which we utilize to perform theoretical analysis. We note that using the doubling dimension is the only way to give theoretical guarantees for popular practical nearest neighbor algorithms such as DiskANN (Indyk & Xu (2023)). Our experiment shows that, regardless of the intrinsic dimensionality, our method achieves consistent improvements.
>
> Regarding the d < D < c*d requirement, we have extensively discussed its validity in our paper. Please refer to lines 419-426 for the discussion and Figure 11 for the empirical D/d values across 15 datasets.
> We acknowledge that selecting datasets with arbitrarily varying D/d values is impractical. The best we can do is test on all popular datasets and report their empirical D/d values for the readers' reference.
>
> Further discussion is appreciated if the reviewer still has any concerns.

---

### Official Review · Reviewer_SVUv · 2024-11-03

**Soundness:** 3
**Presentation:** 2
**Contribution:** 2
**Rating:** 3
**Confidence:** 4

**Summary:**

This paper aims to build a nearest neighbors index over a cheap-to-compute metric $d$, which approximates nearest neighbor search over an expensive, but more accurate, metric $D$. In practice, this means the paper aims to make retrieval over embeddings produced by a small, cheap model emulate retrieval over embeddings produced by a much more expensive model.

The paper proposes using the cheap $d$ to create the index, and then starting each query's search with $d$ before transitioning to using $D$ in distance computations (albeit still leveraging the same index and graph connectivity, which was created on $d$).

**Strengths:**

Extensive suite of datasets and models are benchmarked.
Theory on using a graph built on one metric to navigate a related metric is novel.

**Weaknesses:**

1. Contrived and unrealistic setup: critical to the paper's thesis is the belief that the embedding for $x\in X$ is computed every time the ANN algorithm accesses $x$. It therefore implies that using a cheaper embedding model makes queries cheaper, because these embeddings are re-computed for every query. This is unrealistic because in practice, retrieval systems generally embed all $x\in X$ offline during the indexing phase, so model size has no impact on query latency or throughput. Embedding the entire dataset offline is far more practical because it makes embedding a fixed cost, rather than one linear in the number of queries. For example, if each query involved 1000 embedding calls, then just 1000 queries would lead to the same embedding cost as embedding 1M elements offline. Furthermore, offline indexing can often achieve >10x throughput gains from batching, which makes the computation much more amenable to ML accelerator hardware. This tilts the decision even further in favor of embedding during indexing.
1. Main result from Theorem 3.4 result has a poorly described connection to your proposed metric in the experiments, and the choice of $Q/2$ seems quite arbitrary and isn't explored until the end of the paper (but this isn't described in Section 4.1).
1. Figures are poorly described: "Single metric (limit)" is not described anywhere. I'm assuming the x-axis is $Q$, but $Q$ isn't in the x-axis label despite being extensively referenced in the body text, and I have to infer.
1. Poorly structured: for example, Algorithm 1 is mentioned casually several times in Section 3, but only appears in the appendix, with no indication it could be found there. Theorem 3.4 is the main theorem in the section but is not formatted to emphasize its importance.

**Questions:**

See weaknesses.

---

> ### Author Response · Authors · 2024-11-22
> **Response to Reviewer SVUv**
>
> Thank you for your feedback. We hope that we can clarify a few points.
>
> > Contrived and unrealistic setup: “critical to the paper's thesis is the belief that the embedding for x∈X is computed every time the ANN algorithm accesses x. It therefore implies that using a cheaper embedding model makes queries cheaper, because these embeddings are re-computed for every query. This is unrealistic”
>
> We would like to clarify that computing the embeddings during the query phase is not, in fact,  critical to the paper’s thesis. Instead, it is just a particular instantiation of our framework that we use in our experiments (which allows us to get the state of the art benchmarks on MTEB with very few calls to a large model). As per the discussions in the introduction, there are many other instantiations  of this framework, for example a setting where the metric D( ) itself is computed using a cross-encoder, not using embeddings.  Another instantiation, suggested by Reviewer Dn4i, is a setting where d() is an approximation computed by “sketching” the ground-truth metric D. Our framework has implications to all of those settings.
>
> > Connection between theorem 3.4 and experiments
>
> Theorem 3.4 gives theoretical guarantees for the algorithm used in our experiments (DiskANN). It mathematically proves that a sublinear number (in the dataset) of queries are made to the expensive metric when finding the approximate nearest neighbor, with respect to the expensive metric.The algorithm in the theorem makes no calls to the cheap metric d.
>
> > Description of Figures
>
> We explain the meaning of different methods (bi-metric: our method, bi-metric baseline, and single metric) at the beginning of Section 4.1. The meaning of “single metric (limit)” is straightforward: it represents the maximum accuracy the “single metric” method can achieve if the x-value (i.e., the number of expensive distance computations, which is denoted by Q in the end of page 6) is infinitely increased. This value serves as the upper bound for the NDCG@10 score that any retrieval algorithm can achieve, given the embedding model. Although we believe the meaning is quite clear, we will adopt the reviewer’s suggestion and emphasize this point in the final version of our paper. As mentioned in the end of page 6, Q denotes the “query budget”, which is also the “# of D distance calls” in the figure. We will make this notation more clear in the future version.
>
> > Mention of algorithm 1
>
> We want to emphasize that our contribution is the proposal of a new “bi-metric” framework, but not a new algorithm. Please see our framework from line 70 to 80 and our method from line 320 - 326. Algorithm 1 is the standard greedy search algorithm which we include in the appendix for the sake of completeness. We will emphasize the importance of Theorem 3.4 in the future.

---

> > ### Comment · Reviewer_SVUv · 2024-11-26
> >
> > Using the proposed method for a cross encoder would make more sense. Why weren't any benchmarks done on dual encoder $d$ + cross encoder $D$?

---

> > > ### Author Response · Authors · 2024-11-27
> > >
> > > We argue that the consensus in information retrieval prior to our paper is as follows:
> > >
> > > 1. Efficient search can only be done with distance metrics (the output from a cross-encoder does not form a metric)
> > >
> > > 2. The metric used to build an efficient search index must be the same as the metric used to search the query (or its quantized version).
> > >
> > > Our framework makes significant progress on point 2 by enabling the possibility of building with one metric and testing on another. Whether point 1 can be solved is an interesting question that requires further investigation.
> > >
> > > Due to limitations 1 and 2, the standard practice for using cross-encoders is to first search for the top K (e.g., K=100-1000) candidates according to embedding models, and then rerank these top K candidates with a cross-encoder via a linear scan. This is the “Bi-metric (baseline)” we compare against in Figures 1, 4, 5, 6, 7, 8, 9, and 10 (for different model/algorithm/metric setups in the experiments). In our experiments, we use an expensive embedding model to serve the function of a cross-encoder for the following reasons:
> > >
> > > 1. The main contribution of our work is the proposal of a bi-metric framework. Ideally, we can apply any pair of cheap/expensive distance functions within our framework. It doesn’t matter whether the expensive model is a sentence embedding model or a cross-encoder. However, our theoretical analysis requires that both functions be “distance metrics,” whereas the output from a cross-encoder is not.
> > >
> > > 2. Among the cross-encoders we have tried (e.g., BGE-Reranker, Cohere Reranker), their performance is not stable when the given documents are far from the query. This leads to a strange phenomenon: the more candidates (e.g., when K > 100) are sent to the reranker, the lower the final retrieval result becomes. We hypothesize that the current rerankers are designed to differentiate between close query-document pairs but are not stable enough to provide accurate scores for all query-document pairs.
> > >
> > > Thank you again for your question. We are happy to discuss further if you have any concerns.

---

### Official Review · Reviewer_Dn4i · 2024-11-04

**Soundness:** 3
**Presentation:** 3
**Contribution:** 2
**Rating:** 6
**Confidence:** 3

**Summary:**

This paper proposes a bi-metric framework for nearest neighbor search. It well abstracts a widely used technique for nearest neighbor search, but with several assumptions. The experimental results demonstrate its effectiveness.

**Strengths:**

S1. This paper studies an important problem.

S2. The method proposed seems sound.

S3. The experimental results are good to support the effectiveness.

**Weaknesses:**

W1. The main contribution (Tm 3.4) is built on top of existing works, which makes the novelty insufficient.

W2. The method is applicable to the case where two embedding models are generated for the same data source. However, there still exist another case where a simple embedding model is generated for another expensive embedding for the sake of nearest neighbor search. For example, product quantization for high-dimensional vectors in DiskANN.

W3. The method assumes that d(x, y) ≤ D(x, y) ≤ C · d(x, y), which may not be practical.

**Questions:**

Q1. Could it be possible to extend the results on PQ vs high-dimensional vectors ?

---

> ### Author Response · Authors · 2024-11-22
> **Response to Reviewer Dn4i**
>
> Thank you for your feedback. We hope that we can clarify a few points.
>
> > W1. The main contribution (Tm 3.4) is built on top of existing works, which makes the novelty insufficient.
>
> Theorem 3.4 provides a general framework for using a class of nearest neighbor search algorithms within the two metric framework.Thus,the fact that it applies to multiple existing works is a plus, in our opinion, as it demonstrates that the framework is broadly applicable.
>
> >  The method is applicable to the case where two embedding models are generated for the same data source. However, there still exist another case where a simple embedding model is generated for another expensive embedding for the sake of nearest neighbor search. For example, product quantization for high-dimensional vectors in DiskANN.
>
> > Q1. Could it be possible to extend the results on PQ vs high-dimensional vectors ?
>
>
> Our framework applies to any setting which involves two metrics d and D, such that d approximates D up to a bounded distortion C. Thus, any quantization method that approximately preserves the distances (for example, priority quantization or the method of Chen et al 2023)  can be integrated with our framework, at least in principle.
>
> > The assumption d< D < c*d may be not practical :
>
> As we mention in the footnote on page 5, we have provided an *extensive* empirical analysis to validate this assumption at the end of Section 4.2. In particular,  we verify that 99.9% of randomly sampled $C=D/d$ values from the HotpotQA dataset satisfy $0.6<C<1.5$.. Please refer to Figure 11 for detailed empirical distributions of $C=D/d$ across our 15 datasets. We hope the reviewers will reconsider the validity of our assumption in light of our empirical evidence. We are also happy to discuss any alternative assumptions if suggested by the reviewers.

---

### Author Response · Authors · 2024-11-22
**Thank you to all reviewers**

We thank the reviewers for their valuable feedback. Answers are given in a response to each review.

---

### Note · Authors · 2024-12-02

**Comment:**

Thank you to all reviewers for their valuable feedback.

**Withdrawal Confirmation:**

I have read and agree with the venue's withdrawal policy on behalf of myself and my co-authors.